# Fitness variation across subtle environmental perturbations reveals local modularity and global pleiotropy of adaptation

**Grant Kinsler[1†], Kerry Geiler-Samerotte[1,2†], Dmitri A Petrov[1]\***

[1]Department of Biology, Stanford University, Stanford, United States; [2]Center for Mechanisms of Evolution, School of Life Sciences, Arizona State University, Tempe, United States

**Abstract** Building a genotype-phenotype-fitness map of adaptation is a central goal in evolutionary biology. It is difficult even when adaptive mutations are known because it is hard to enumerate which phenotypes make these mutations adaptive. We address this problem by first quantifying how the fitness of hundreds of adaptive yeast mutants responds to subtle environmental shifts. We then model the number of phenotypes these mutations collectively influence by decomposing these patterns of fitness variation. We find that a small number of inferred phenotypes can predict fitness of the adaptive mutations near their original glucose-limited evolution condition. Importantly, inferred phenotypes that matter little to fitness at or near the evolution condition can matter strongly in distant environments. This suggests that adaptive mutations are locally modular — affecting a small number of phenotypes that matter to fitness in the environment where they evolved — yet globally pleiotropic — affecting additional phenotypes that may reduce or improve fitness in new environments.

**\*For correspondence:**
dpetrov@stanford.edu

[†]These authors contributed equally to this work

**Competing interests:** The authors declare that no competing interests exist.

## Introduction

Laboratory evolution experiments are opening an unprecedented window into the dynamics and genetic basis of adaptive change by de novo mutation (*Crozat et al., 2010*; *Good et al., 2017*; *Huang et al., 2018*; *Lang et al., 2013*; *Levy et al., 2015*; *Tenaillon et al., 2012*; *Venkataram et al., 2016a*). One of the key insights revealed by these studies is that in many systems, evolution can initially proceed rapidly via many large-effect single mutations. While the identities of these adaptive mutations are often unique to a specific replicate of the evolutionary experiment, across many replicates they tend to occur in similar functional units (e.g. genes and pathways) (*Crozat et al., 2010*; *Fumasoni and Murray, 2020*; *Good et al., 2017*; *Huang et al., 2018*; *Lang et al., 2013*; *Levy et al., 2015*; *Tenaillon et al., 2012*; *Venkataram et al., 2019*, *Venkataram et al., 2016a*). Thus, although the diversity of mutations suggests that there might be many ways to adapt, the much smaller number of apparent functional units implies, in contrast, that most adaptive mutations affect a small set of key phenotypes (*Figure 1A*).

Consider the seminal study by *Tenaillon et al., 2012* in which 115 populations were evolved at high temperature for ~2000 generations. While the authors identified over a thousand mutations that were largely unique to each population, the number of affected genes was much smaller with 12 genes being hit over 25 times each. Even greater convergence was seen at higher levels of organization such as operons. Similarly, *Venkataram et al., 2016a* found that, of the hundreds of unique genetic mutations that occur during adaptation to glucose-limitation, the vast majority fall into a relatively small number of genes (mostly *IRA1*, *IRA2*, *GPB2*, *PDE2*) and primarily two pathways — Ras/

**eLife digest** One of the goals of evolutionary biology is to understand the relationship between genotype, phenotype, and fitness. An organism's genes – its genotype – determine its physical and behavioral traits – its phenotype. Phenotypes, in turn, affect the organisms' chances of survival and reproduction – its fitness. However, mapping the relationships among these three variables is far from easy. Recently researchers have become able to identify many genetic mutations that increase an organism's fitness, but it is more difficult to work out how these mutations affect an organism's phenotype, and why they are beneficial.

The mutations that help organisms thrive in a particular environment are often limited to a handful of genes that affect similar biological processes. For example, microbes that grow in environments with limited sugar tend to accumulate mutations in genes involved in systems that determine whether to grow fast and carelessly or to be careful in case the sugar is never replenished. It is possible that these mutations all affect the same one or two phenotypes, such as the decision to grow or to hunker down. If this were the case, researchers should be able to easily predict how well these organisms adapt to new environments. However, it is possible that specific mutations affect several phenotypes, but these extra effects remain invisible until the environment changes and these phenotypes are revealed.

To explore this possibility, Kinsler, Geiler-Samerotte, and Petrov obtained hundreds of individual yeast strains that each contained a different mutation that improved the yeast's fitness in a low sugar environment. They placed these strains into similar environments and measured their fitness. The patterns observed were used to build several models that predicted how many phenotypes each mutation must affect to explain the changes in fitness.

Kinsler, Geiler-Samerotte and Petrov found that the model in which only five phenotypes were affected by the mutations was able to predict the fitness of the yeast in low-sugar environments. However, to predict the fitness of the same mutations in environments that were very different, the model had to include eight phenotypes. This suggests that although the mutations that helped yeast do well in the low sugar environment were similar in their benefits in this environment, they were not truly all the same. In fact, some mutations were quite different from the others in terms of their hidden phenotypic effects.

The hidden effects of mutations can be positive or negative. One mutation might cause an organism to die in a new environment, whereas another might allow it to thrive. Understanding how this works has implications not only for evolutionary biology, but also for medical research. Pathogens that cause infection, and cells that cause cancer, often accumulate mutations in small numbers of crucial genes. Understanding how these mutations affect phenotypes that become important as the environment changes – for instance as the cells encounter new challenges as a tumor grows – and whether different mutations have different hidden effects, could improve treatments in the future.

PKA and TOR/Sch9. Thus, despite the diversity of mutations, it is possible that all their effects can be mapped in one or few dimensions required to describe their effects on the Ras/PKA or TOR/Sch9 pathways. These are just two examples, but the pattern has been seen repeatedly (*Barghi et al., 2019*; *Crozat et al., 2010*; *Good et al., 2017*; *Lang et al., 2013*; *Lind et al., 2015*). Note that this pattern is seen not only in experimental evolution but also in cancer evolution. Individual tumors are largely unique in terms of specific mutations, but these mutations affect a much smaller set of driver genes and an even smaller number of higher functional units such as signaling pathways (*Bailey et al., 2018*; *Hanahan and Weinberg, 2011*; *Hanahan and Weinberg, 2000*; *Sanchez-Vega et al., 2018*; *Sondka et al., 2018*).

The mapping of adaptive mutations to a smaller number of functional units and thus a low-dimensional space representing the small number of phenotypes that they collectively affect (*Figure 1A*) is consistent with theoretical models of adaptation. These theoretical models argue that adaptive mutations, especially those of substantial fitness benefit, cannot affect too many phenotypes at once as most such effects should be deleterious and thus inconsistent with the overall positive effect on fitness (*Fisher, 1930*; *Orr, 2000*). More recent studies likewise suggest that selection against

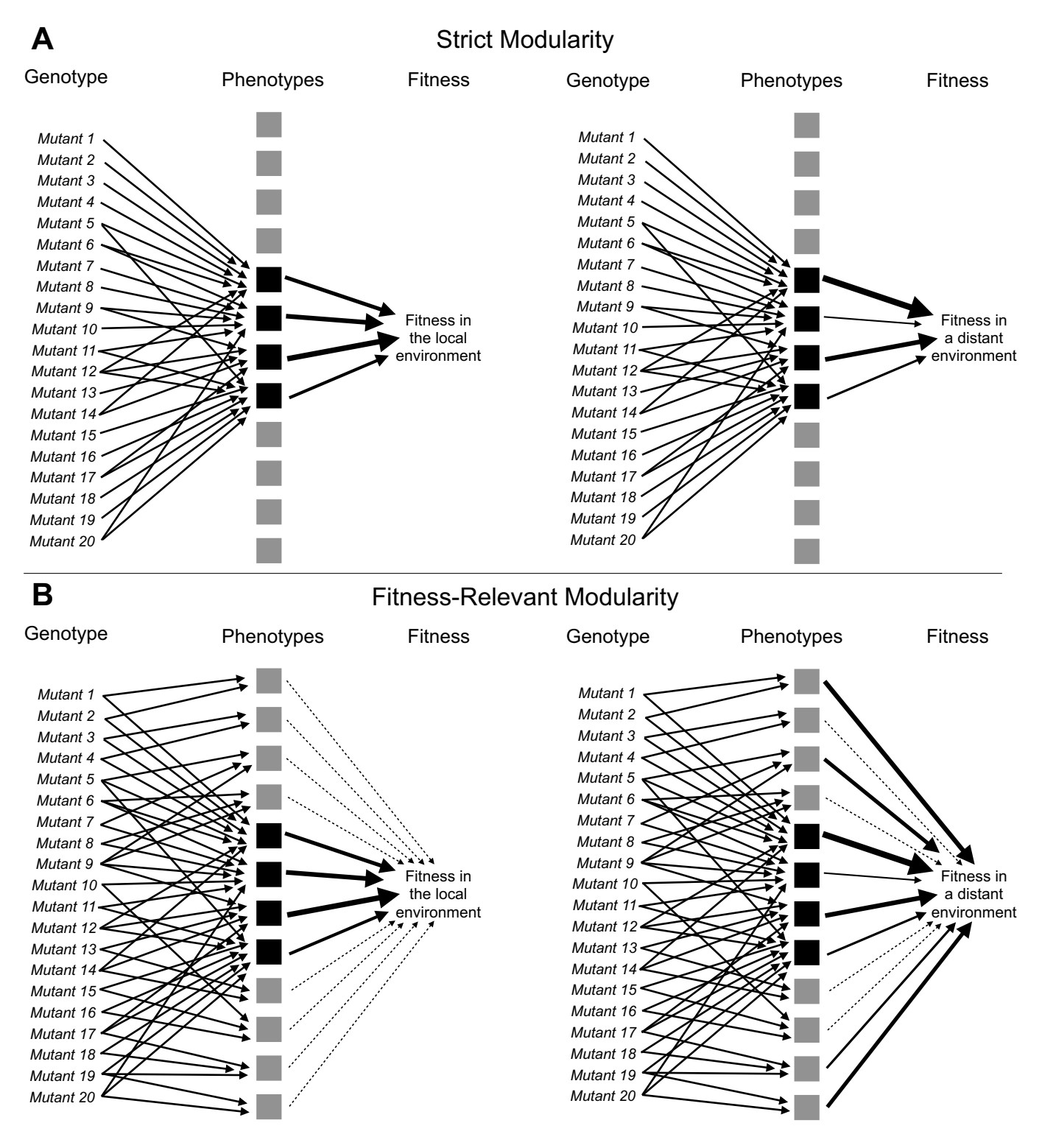

**Figure 1.** Adaptive mutations can be locally modular and globally pleiotropic. (**A**) In the 'strict modularity' model, a collection of adaptive mutations may affect a small number of phenotypes (four black squares). If these adaptive mutations only affect these phenotypes then fitness in both the environment they evolved in (local environment) and other environments (distant environment) is determined solely by these phenotypes. (**B**) Alternatively, in the 'fitness-relevant modularity' model, these mutations may collectively (and individually) affect many phenotypes, but only a small number of phenotypes may matter to fitness in the local environment (those indicated by black squares with thick arrows pointing to fitness), whereas other phenotypes may make very small contributions to fitness (those indicated by the gray squares and thin, dashed lines leading to fitness). Under

*Figure 1 continued on next page*

*Figure 1 continued*

this model, the contribution of each phenotype to fitness can change depending on the environment. Thus, fitness differences between mutants that behave similarly in the local environment can be revealed by measuring fitness in more distant environments. Such fitness differences reveal the presence of phenotypic differences between mutants.

mutations with high pleiotropy, that is mutations that affect many phenotypes, has resulted in a modular architecture of the genotype-phenotype map, in which genetic changes can influence some phenotypes without disturbing others (*Altenberg, 2005*; *Collet et al., 2018*; *Hartwell et al., 1999*; *Melo et al., 2016*; *Wagner et al., 2007*; *Wagner and Altenberg, 1996*; *Wagner and Zhang, 2011*; *Welch and Waxman, 2003*). This architecture would allow single mutations to have a large effect on a small number of important phenotypes. It would also explain the observation that even very large collections of mutations that provide a fitness benefit in a particular condition are not diverse in terms of affected genes, pathways, and phenotypes. The reason for this is that only mutations that affect the genes, pathways, and phenotypes corresponding to the module most relevant to adaptation in that condition will be observed. We term this model in which mutations only affect a small number of phenotypes 'strict modularity'.

While theoretically appealing, the possibility that observed adaptive mutations indeed affect only a very small number of phenotypes is difficult to reconcile with the notion that organisms are tightly integrated (*Kacser and Burns, 1981*; *Paaby and Rockman, 2013*; *Rockman, 2012*). Further, there is experimental evidence of widespread pleiotropy, for example, from genome-wide association studies that suggest that every gene can influence every trait, at least to some extent (*Boyle et al., 2017*; *Chesmore et al., 2018*; *Sella and Barton, 2019*; *Sivakumaran et al., 2011*; *Visscher and Yang, 2016*). It is possible that pleiotropy is common, but strongly adaptive mutations observed in experimental evolution are unusual in that they have few phenotypic effects. Another possibility is that these mutations do have pleiotropic side effects, but these matter little to fitness in the condition where these mutants evolved (*Figure 1B*, left side). We term this model 'fitness-relevant modularity' because these mutations are not strictly modular with respect to all the phenotypes they affect, but they are effectively modular because only a subset of these phenotypes are relevant to fitness in the evolution condition. Here, we do not need to claim that these phenotypic effects *never* matter to fitness but rather that they do not matter substantially to fitness in the condition where they evolved. In fact, the key prediction of this model is that one should be able to detect latent pleiotropy and reveal the additional phenotypic effects of these mutants by demonstrating their varied fitness consequences in other conditions or environments (*Figure 1B*, right side). Note that we cannot test this prediction by demonstrating antagonistic pleiotropy, that is that mutations that are adaptive in one environment have fitness tradeoffs in other environments (*Dillon et al., 2016*; *Jerison et al., 2020*). Antagonistic pleiotropy could indeed indicate that the mutations affect many phenotypes, some of which only hinder fitness in certain environments. But it could also indicate that the adaptive mutations all change the same phenotype in a way that improves fitness in some environments and hinders fitness in others.

If the 'fitness-relevant modularity' model depicted in *Figure 1B* is true then it is possible that adaptive mutations are *locally modular* — that they affect very few phenotypes that matter to fitness in the evolution condition — and *globally pleiotropic*. Under this model, the large number of distinct mutations available to adaptation becomes important. Indeed while these mutations tend to influence similar genes and pathways, their phenotypic effects do not simply collapse to a low-dimensional space. Instead, this genetic diversity becomes a source of consequential phenotypic diversity, but only once these genetic variants leave the local environment in which they originated.

In order to test this model and better understand the genotype-phenotype-fitness map, we face the difficult task of identifying which phenotypes are affected by the adaptive mutations and then determining how these phenotypes contribute to fitness. This is a challenging problem as the possible number of phenotypes one can measure is effectively infinite, for example the expression level of every gene or the quantity of every metabolite (*Coombes et al., 2019*; *Mehlhoff et al., 2020*). Further, many measurable phenotypes are related in complex ways (*Geiler-Samerotte et al., 2020*). Mapping their contribution to fitness requires a complete understanding of how genetic changes lead to molecular changes and how these percolate to higher functional levels and ultimately influence fitness (*Kemble et al., 2020*). This might be possible to do in some cases where the phenotype

to fitness mapping is simple (e.g. antibiotic resistance driven by a specific enzyme or tRNA or protein folding mediating specific RNA or protein function; *Baeza-Centurion et al., 2019*; *Cowperthwaite et al., 2005*; *Diss and Lehner, 2018*; *Domingo et al., 2019*; *Harmand et al., 2017*; *Karageorgi et al., 2019*; *Li and Zhang, 2018*; *Otwinowski et al., 2018*; *Pressman et al., 2019*; *Sarkisyan et al., 2016*; *Starr et al., 2018*; *Weinreich, 2006*) but is exceptionally difficult for complex phenotypes. In the case of the adaptive mutations from *Venkataram et al., 2016a* mentioned above, we might be able to use our knowledge of the Ras/PKA pathway to make a guess about what phenotypes they affect. We know that many of these mutations result in the loss of negative regulators of the Ras/PKA pathway (*IRA1*, *IRA2*, *GPB2*, *PDE2*). Thus, we might guess that these adaptive mutations all lead to an increase in the amount of active PKA. Then we could use more traditional approaches to confirm this hypothesis, for example, by measuring the levels of PKA through functional assays. However, even if these mutations do increase PKA activity, it is not clear how this effect percolates through the system, or what other phenotypic effects we might miss by using such a directed approach to investigate the genotype-phenotype-fitness map.

Moreover, to distinguish between the model in which mutations affect a small number of phenotypes ('strict modularity' as shown in *Figure 1A*) and the model in which mutations affect many phenotypes, albeit with few contributing substantially to fitness in the evolution condition ('fitness-relevant modularity' as shown in *Figure 1B*), we need to understand these genotype-phenotype-fitness maps not only in the environment in which adaptive mutants evolved but also in other environments. And we need to do this for many adaptive mutants so that we can assess the extent to which different mutants affect different phenotypes. Considering the scope of this challenge, it is not surprising that despite much theoretical discussion of modularity and pleiotropy as it relates to adaptation, experimental approaches to address these questions have lagged behind.

Here, we suggest a way to model the genotype-phenotype-fitness relationship that avoids the problem of measuring each phenotype and its effect on fitness explicitly. We argue that it is possible to investigate the genotype-phenotype-fitness map by comparing how the fitness effects of many mutations change across a large number of environments. The way each mutant's fitness varies across environments must be related to its phenotype, and thus the way mutants co-vary in fitness across environments tells us whether they affect similar fitness-relevant phenotypes. We can use these profiles of fitness across a set of environments to identify the total number of fitness-relevant phenotypes that must be affected across a collection of adaptive mutants, the extent to which different mutants affect different phenotypes, and whether the contribution of each phenotype to fitness changes across environments. Importantly, the phenotypes we identify with this approach are abstract entities rather than measured cell properties. Nevertheless, these abstract phenotypes reflect the causal effects of adaptive mutations on fitness.

Here, we build a genotype-(abstract)phenotype-fitness model for hundreds of adaptive yeast mutants that originally evolved in a glucose-limited environment. We use this model to accurately predict the fitness of these mutants across a set of 45 environments that vary in their similarity to the evolution condition. We find that the fitness behavior of adaptive mutations near the evolution condition can be described by a low-dimensional phenotypic model. In other words, these mutants affect a small number of phenotypes that matter to fitness in the glucose-limited condition in which they evolved. We find that this low-dimensional phenotypic model makes accurate predictions of mutant fitness in novel environments even when they are dissimilar to the evolution condition. Moreover, we find that some phenotypes that contribute very little to fitness in the evolution condition become surprisingly important in some novel environments. This suggests that adaptive mutations are globally pleiotropic in that they affect many phenotypes overall, but that they are locally modular in that only a small number of these phenotypes have substantial effects on fitness in the environment they evolved in. Overall, we suggest that this set of adaptive mutations contains substantial and consequential latent phenotypic diversity, meaning that despite targeting similar genes and pathways, different adaptive mutants may respond differently to future evolutionary challenges. This finding has important consequences for understanding how directional selection can generate consequential phenotypic heterogeneity both in natural populations and also in the context of diseases, such as cancer and viral or bacterial infections. In addition, our results show that our abstract, top-down approach is a promising route of analysis for investigating the phenotypic and fitness consequences of mutation.

## Results

### Mutants that improve fitness under glucose limitation vary in their genotype-by-environment interactions

A previous evolution experiment generated a collection of hundreds of adaptive yeast mutants, each of which typically harbors a single independent mutation that provides a benefit to growth in a glucose-limited environment (*Levy et al., 2015*). Many of these mutants, which began the evolution experiment as haploids, underwent whole-genome duplication to become diploid, which improved their relative fitness (*Venkataram et al., 2016a*). Some of these diploids acquired additional mutations, including increased copy number of either chromosome 11 or 12 as well as point mutations, which generated additional fitness benefits. The adaptive mutants that remained haploid acquired both gain- and loss-of-function mutations in nutrient-response pathways (Ras/PKA and TOR/Sch9). Some other mutations were also observed, including a mutation in the HOG pathway gene *SSK2* (*Venkataram et al., 2016a*). Although these mutants have been well-characterized at the level of genotype and fitness, it is unclear what phenotypes they affect. The first question we address is whether these diverse mutations collectively affect a large number of phenotypes that matter to fitness, or whether these mutants are functionally similar in that they collectively alter a small set of fitness-relevant phenotypes.

Understanding the map from genotype to phenotype to fitness is extremely challenging because each genetic change can influence multiple traits, not all of which are independent or contribute to fitness in a meaningful way. We contend with this challenge by measuring how the relative fitness of each adaptive mutant changes across a large collection of similar and dissimilar environments, which we term the 'fitness profile'. When a group of mutants demonstrate similar responses to environmental change, we conclude that these mutants affect similar phenotypes. By clustering mutants with similar fitness profiles across a collection of environments, we can learn about which mutants influence similar phenotypes, as well as estimate the total number of fitness-relevant phenotypes represented across all mutants in all investigated environments.

Because our mutant strains are barcoded, we can use previously established methods to measure their relative fitness in bulk and with high precision (*Venkataram et al., 2016a*). Specifically, we compete a pool of the barcoded mutants against an ancestral reference strain over the course of several serial dilution cycles. During each 48 hr cycle, the yeast are given fresh glucose-limited media which supports eight generations of exponential growth after which glucose is depleted and cells transition to non-fermentable carbon sources. After every 48 hr cycle, we transfer ~$5 \times 10^7$ cells to fresh media to continue the growth competition. We also extract DNA from the remaining cells to PCR amplify and sequence their barcodes. We repeat this process four times, giving us an estimate of the frequency of each barcode at five time-points. By quantifying the log-linear changes in each barcode's frequency over time and correcting for the mean-fitness change of population, we can calculate the fitness of each barcoded mutant relative to the reference strain (*Figure 2A*; Materials and methods).

Using this method, we quantify the fitness of a large number of adaptive mutants in 45 environments. We focus on a set of 292 adaptive mutants that have been sequenced, show clear adaptive effects in the glucose-limited condition in which these mutants evolved (hereafter 'evolution condition'; EC) (*Figure 2B*; *Supplementary file 1*), and for which we obtained high-precision fitness measurements in all 45 environments. These environments include some experiments from previously published work (*Li et al., 2018*; *Venkataram et al., 2016a*), as well as 32 new environments including replicates of the evolution condition, subtle shifts to the amount of glucose, changes to the shape of the culturing flask, changes to the carbon source, and addition of stressors such as drugs or high salt (*Supplementary file 2*).

In order to determine the total number of phenotypes that are relevant to fitness in the EC, we focus on environments that are very similar to the EC but still induce small yet detectable perturbations in fitness. We do so because the phenotypes that are the most relevant to fitness may change with the environment (*Figure 1B*). Thus, we partition the 45 environments into a set of 'subtle' perturbations, from which we will detect the phenotypes relevant to fitness near the EC, and 'strong' perturbations which we will use to study whether these mutants influence additional phenotypes that matter in other environments (*Figure 1B*).

To partition environments into subtle and strong perturbations of the EC, we rely on the nested structure of replicate experiments performed in the EC. We assayed fitness in the EC on nine

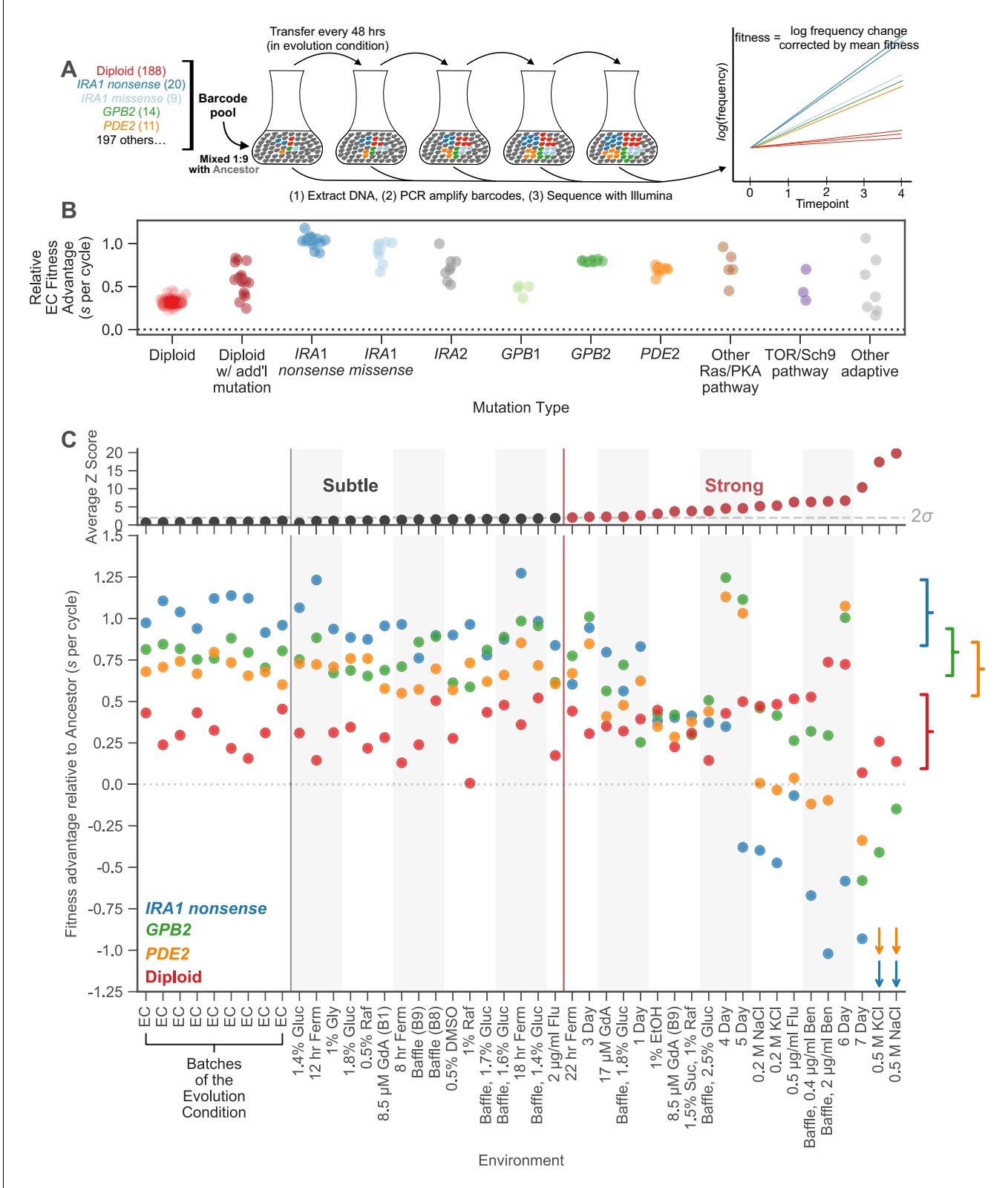

**Figure 2.** Measuring fitness for a collection of adaptive mutants across many environments reveals gene-by-environment interactions. (**A**) Schematic of fitness measurement procedure. Adaptive mutants tagged with DNA barcodes are pooled at a 1:9 ratio with an ancestral reference strain. The pool is then propagated for several growth cycles, where the population is diluted into fresh media at fixed time intervals. DNA is extracted from each time-point, and the barcode region is PCR amplified and then sequenced. A mutant's relative fitness is calculated based on the rate of change of its

*Figure 2 continued on next page*

*Figure 2 continued*

barcode's frequency, corrected for the mean fitness of the population (see Materials and methods). Relative fitness is calculated in units of 'per cycle', representing the improvement of each barcode relative to the reference over the course of the time between transfers. (B) Fitness advantage of each mutant in the evolution condition relative to the ancestor. This fitness advantage is measured per transfer cycle and calculated as the average across all nine Evolution Condition (EC) batches. (C) (top) Environments are ordered from left to right depending on the degree to which they perturb mutant fitness from the average fitness observed across all EC batches. Environments in which average mutant fitness is within two standard deviations of average mutant fitness across EC batches are denoted in black and make up the subtle perturbation set. Environments in which aggregate mutant behavior exceeds two standard deviations are shown in red and make up the strong perturbations set. (bottom) This plot displays, for the four most common types of adaptive mutation observed in response to glucose limitation (*Venkataram et al., 2016a*), the average fitness in each of the 45 environments we study. Brackets on the right represent the amount of variation in fitness observed for each type of mutation across the EC batches, with the notch representing the mean and the arms representing two standard deviations on either side of the mean. For visualization purposes, we represent relative fitness values below −1.25 as arrows. Specifically, *PDE2* mutants (orange arrows) have on average fitness −3.3 and −3.4 in 0.5 M KCl and 0.5 M NaCl, respectively. *IRA1 nonsense* mutants (blue arrows) have an average fitness −3.0 and −4.2 in 0.5 M KCl and 0.5 M NaCl, respectively. The online version of this article includes the following source data and figure supplement(s) for figure 2:

**Source data 1.** Fitness measurement data.
**Figure supplement 1.** Noise model is a conservative measure of uncertainty.
**Figure supplement 2.** Replicates show consistent estimates of fitness.

different occasions which we term 'batches'. Each batch contained multiple replicates. We observe much less variation across replicates than across batches (p<1e-5 from permutation test). Variation across batches likely reflects environmental variability that we were unable to control (e.g. slight fluctuations in incubation temperature due to limits on the precision of the instrument, slight differences in the media reflective of the limits on the precision of our scale). These differences between batches are as subtle as possible in our experimental setup, as they represent the limit of our ability to minimize environmental variation. Thus, variation in fitness across the EC batches serves as a natural benchmark for the strength of other environmental perturbations. If the deviations in fitness caused by an environmental perturbation are substantially stronger than those observed across the EC batches, we call that perturbation 'strong'.

More explicitly, to determine whether a given environmental perturbation is subtle or strong, we subtract the fitness of adaptive mutants in this environment from their average across the EC batches. We then compare this difference to the variation in fitness observed across the EC batches. Sixteen environmental perturbations provoked fitness differences that were similar to those observed across EC batches (Z-score <2). These environments, together with the nine EC batches, make up a set of subtle environmental perturbations. The remaining 20 environments, where the average deviation in fitness is substantially larger than that observed across batches (Z-score >2), were classified as strong environmental perturbations (*Figure 2C*, top; Materials and methods). Note that when we use different subsets of the subtle environmental perturbations, our qualitative conclusions hold, indicating they are not sensitive to our particular choice of which environments to classify as subtle or strong (*Figure 4—figure supplement 1*).

The rank order of the fitnesses of many mutations is largely preserved across the 25 environments that represent subtle perturbations (*Figure 2C*, bottom). For example, *IRA1 nonsense* mutants, which are the most adaptive in the EC, generally remain the most adaptive across the subtle perturbations. Additionally, the *GPB2* and *PDE2* mutants have similar fitness effects across EC batches and only occasionally switch order across the subtle environmental perturbations. In contrast, the 20 environments that represent strong perturbations reveal clear genotype-by-environment interactions (*Figure 2C*, bottom). For example, altering the transfer time from 48 to 24 hr (the '1 Day' environment in *Figure 2C*) affects *GPB2* mutants more strongly compared to the other mutants in the Ras/ PKA pathway, including *IRA1* and *PDE2*. The strongest environmental perturbations reveal clear tradeoffs for some of these adaptive mutants. For example, *PDE2* and *IRA1 nonsense* but not *GPB2* mutants are particularly sensitive to osmotic stress as indicated by the NaCl and KCl environments. Additionally, *IRA1 nonsense* mutants become strongly deleterious in the long transfer conditions that experience stationary phase (5-, 6-, 7-Day environments) (*Li et al., 2018*). In contrast to complex behavior exhibited by the adaptive haploids, the diploids appear to be relatively robust to strong tradeoffs, appearing similarly adaptive across all perturbations, subtle and strong.

The observation that different mutants have different and fairly complex fitness profiles suggests that they have different phenotypic effects. Even *PDE2* and *GPB2,* which have similar fitnesses in the EC and are negative regulators of the same signalling pathway, have different fitness profiles. Do these diverse phenotypic effects contribute to fitness in the EC? To examine how many phenotypes matter to fitness in the EC, we test whether it is possible to create low-dimensional models that capture the complexity of the fitness profiles of all adaptive mutants across all subtle perturbations.

## A model including eight fitness-relevant phenotypes captures fitness variation across subtle environmental perturbations

We utilize these complex fitness profiles to estimate the number of phenotypes that contribute to fitness in the EC. Given that many of these mutants affect genes in the same nutrient response pathway, the number of unique phenotypes they affect may be small. Alternatively, given the observation that these mutants have different interactions with environments that represent strong perturbations (*Figure 2C*), this number may be large. We use singular value decomposition (SVD) to ask how much of the complexity in these fitness profiles can be captured by a low-dimensional phenotypic model (*Figure 3A*). SVD is a dimensionality reduction approach which here decomposes fitness profiles into two abstract multi-dimensional spaces described below.

The first space, *P*, represents the phenotypic effects of mutants, where each phenotype is represented as a dimension (there are *k* phenotypic dimensions depicted in *Figure 3A*). Each mutant is represented by coordinates specifying a location in the phenotype space *P* (e.g. mutant one having coordinates $(p_{11}, p_{12}, p_{13}, ..., p_{1k})$). The ancestral reference lineage, which, by definition, has relative fitness zero in every environment, is placed at the origin (e.g. (0, 0, 0, . . . 0)) in this phenotypic space. In this sense, we can think of a mutation's effect on any phenotype as a measure of the distance from the location of the mutant in that phenotypic dimension to the origin.

The second space, *E*, represents the contribution of each of the phenotypes in *P* to fitness, and thus has the same number of dimensions as *P*. If a phenotype does not contribute substantially to fitness in any environment, it is not represented as a dimension in either space. Therefore, our model captures only fitness-relevant phenotypes. In space *E*, each environment is represented by coordinates specifying a location (e.g. environment one having coordinates $(e_{11}, e_{21}, e_{31}, ..., e_{k1})$). These coordinates in *E* reflect the contribution (weight) of each of the *k* phenotypic dimensions on fitness in that environment. For example, an environment where only a single phenotype matters to fitness would be placed at the origin for all the axes, except for the axis corresponding to the single phenotypic dimension that matters. Environments for which the same phenotypes contribute to fitness will be placed closer together in the space *E*.

In this model, each phenotype contributes to fitness independently, by definition, such that the fitness of mutant *i* in environment *j* is determined by each phenotypic effect of mutant *i,* scaled by the contribution of that phenotype to fitness in environment *j*. A linear combination of these weighted phenotypic effects determines the fitness of mutant *i* in environment *j*:

$$f_{ij} = p_{i1}e_{1j} + p_{i2}e_{2j} + p_{i3}e_{3j} + ... + p_{ik}e_{kj}$$

In this model, mutants with similar fitness profiles, for example mutants 1 and 2 in *Figure 3A*, will be inferred as having similar phenotypic effects, and thus be located near each other in the phenotypic space *P*. Mutants with dissimilar fitness profiles, for example mutants 3 and 4 in *Figure 3A*, can be inferred to have at least some differing phenotypic effects, which might be mediated by a different effect on a single phenotypic component or different effects on many. Mutants with dissimilar fitness profiles are informative about the number of dimensions needed in this abstract model of phenotypic space.

This genotype-phenotype-fitness model that we generate using SVD harkens to Fisher's geometric model (FGM), which defines an abstract space of orthogonal phenotypes relevant to fitness (*Fisher, 1930*). Others have utilized FGM to answer questions about the number of phenotypes affected by mutations, although most previous work focuses on deleterious mutations and how their impacts vary across genetic backgrounds rather than environments (*Blanquart et al., 2014*; *Blanquart and Bataillon, 2016*; *Lourenço et al., 2011*; *Martin and Lenormand, 2006*; *Poon and Otto, 2000*; *Tenaillon et al., 2007*; *Weinreich and Knies, 2013*). A key difference between FGM

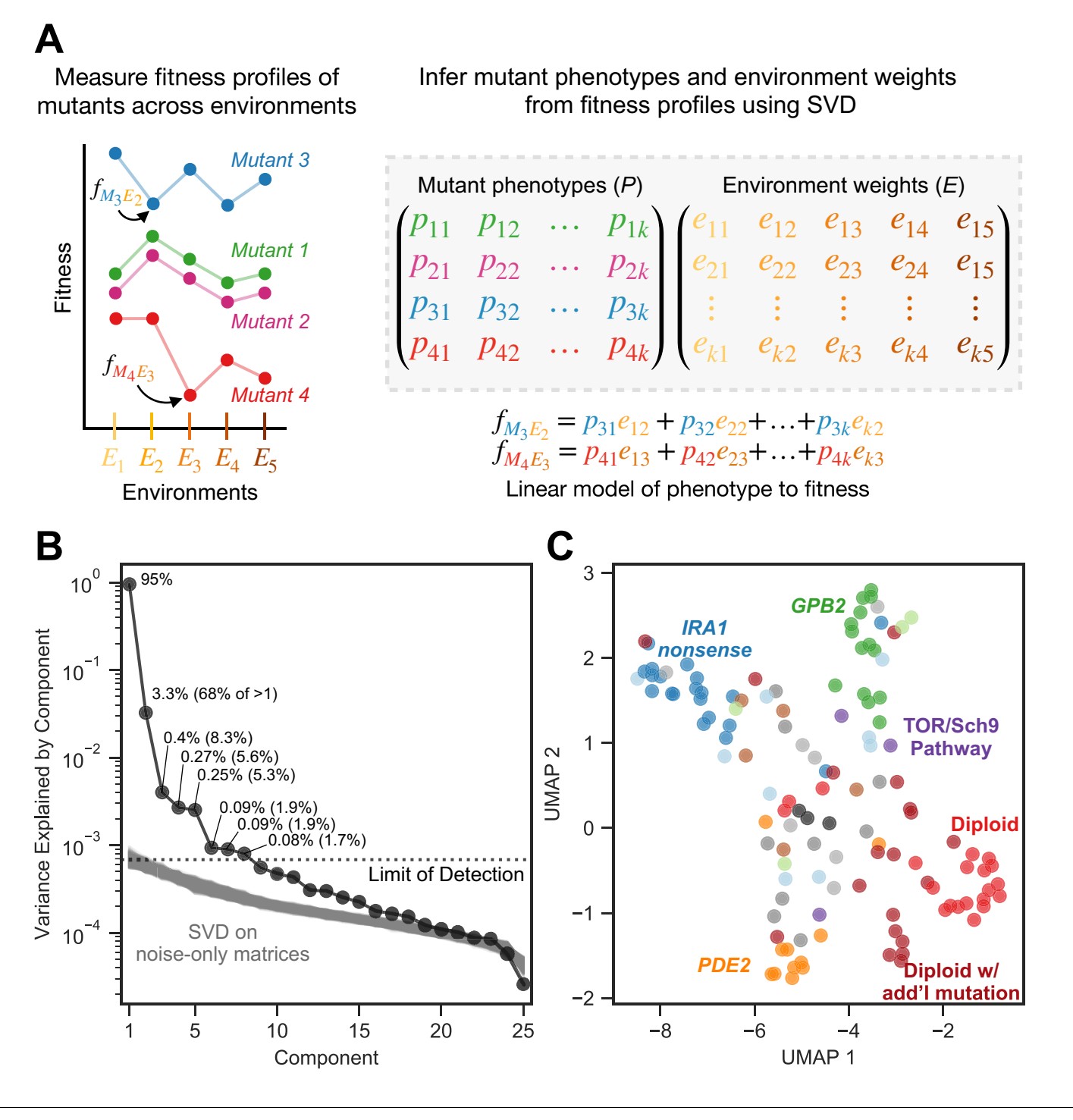

**Figure 3.** Subtle environmental perturbations reveal an eight-component phenotypic model that reflects known biological features. (**A**) To infer fitness-relevant phenotypes, we measure the fitness of mutants in a collection of environments and compare their fitness profiles. Mutants with similar fitness profiles (mutants 1 and 2) are inferred to have similar effects on phenotypes. Mutants with dissimilar fitness profiles (mutants 3 and 4) are inferred to have dissimilar phenotypic effects. We use SVD to decompose these fitness profiles into a model consisting of two abstract spaces: one that represents the fitness-relevant phenotypes affected by mutants (*P*) and another which represents the degree to which each phenotype impacts fitness in each environment (*E*). Here, we represent the model with *k* fitness-relevant phenotypes. The model's estimate for fitness for a particular mutant in a particular environment is a linear combination of each mutant phenotype (mutant one is represented by the vector $(p_{11}, p_{12}, p_{13}, ..., p_{1k})$) scaled by the degree to which that phenotype affects fitness in the relevant environment (environment one is represented by the vector $(e_{11}, e_{12}, e_{13}, ..., e_{1k})$). We show two examples of the equation used to estimate fitness for the mutants and environments highlighted in the left panel. Note that, for presentation purposes,

*Figure 3 continued on next page*

*Figure 3 continued*

we show SVD as inferring two matrices. It in fact infers three, but is consistent with our presentation if you fold the third matrix, which represents the singular values, into *E* (see Materials and methods). (**B**) Decomposing the fitness profiles of 292 adaptive mutants across 25 subtle environmental perturbations reveals eight fitness-relevant phenotypic components. The variance explained by each component is indicated as a percentage of the total variance. The percentages in parentheses indicate the relative amount of variation explained by each component when excluding the first component. Each of these components explain more variation in fitness than do components that capture variation across a simulated dataset in which fitness varies due to measurement noise. These simulations were repeated 1000 times (gray lines) and used to define the limit of detection (dotted line). (**C**) An abstract space containing eight fitness-relevant phenotypic components reflects known biological features. This plot shows the relationships of the mutants in a seven-dimensional phenotypic space that excludes the first component, visualized using Uniform Manifold Approximation and Projection (UMAP). Mutants that are close together have similar fitness profiles and are inferred to have similar effects on fitness-relevant phenotypes. Mutants with mutations in the same gene tend to be closer together than random, in particular *IRA1 nonsense* mutants in dark blue, *GPB2* mutants in dark green, *PDE2* mutants in dark orange, and diploid mutants in red. Six diploid mutants that had higher than average diploid EC fitness (and thus are likely to harbor additional mutation(s) so are categorized as 'diploid with additional mutation') also form a cluster. Colors are as in *Figure 2B*; *IRA1 missense* mutants shown in light blue, *IRA2* in dark gray, *GPB1* in light green, other Ras/PKA pathway mutants in brown, TOR/Sch9 pathway mutants in purple, other adaptive mutants in light gray, and known neutral lineages in black.

The online version of this article includes the following figure supplement(s) for figure 3:

**Figure supplement 1.** Accurate predictions of the number of phenotypic components in simulated data.

**Figure supplement 2.** The first component represents the mean fitness of each mutant in the 25 subtle perturbations, as well as the mean impact of each perturbation on fitness.

**Figure supplement 3.** Locations of mutants and environments in phenotype space.

**Figure supplement 4.** Low-dimensional phenotypic models, and subsets of such models, cluster mutants by gene and mutation type.

and our model is that our model does not make assumptions about the distribution of phenotypic effects or whether the relationship between mutations in phenotype space is additive.

Here, we utilize SVD to count the number of phenotypes that contribute to fitness in the original glucose-limited environment in which these adaptive mutants evolved. We used SVD to build an abstract model that captures fitness profiles of all 292 adaptive mutants across the 25 subtle perturbations. This model suggests that the majority of the variation in fitness for the 292 adaptive mutants across the 25 subtle perturbations can be explained by eight phenotypic dimensions. The first phenotypic component is very large and explains 95% of variation in fitness across all mutants and all subtle perturbations (*Figure 3B*). This component captures the variation in fitness explainable in the absence of genotype-by-environment interactions, where each mutation has a single effect that is scaled by the environment. As such, this first component effectively represents each mutant's average fitness in the EC (*Figure 3—figure supplement 2A*) and the average impact of each subtle perturbation on mutant fitness (*Figure 3—figure supplement 2B*). It is not surprising that this component explains much of this variation, as the fitness of mutants in the EC should be predictive of fitness in similar environments. The next seven components capture additional variation not detectable from the simple one-component model and thus represent genotype-by-environment interactions. Of these, the first four capture 87% of the variation not captured by component one (67.8%, 8.3%, 5.6%, and 5.3%, respectively). The remaining three interaction components each capture less than 2% of the variation not captured by component one (*Figure 3B*). We cannot distinguish any additional components, beyond these eight, from noise. This is because we see components that explain a similar amount of variation when we apply SVD to datasets composed exclusively of values generated by our noise model (*Figure 3B*; see Materials and methods and *Figure 3—figure supplement 1* for additional details).

We confirm that these eight phenotypic components capture meaningful biological variation in fitness by using bi-cross-validation. Specifically, we designate a balanced set of 60 of the 292 mutants as a training set, chosen such that the recurrent mutation types — diploids, high-fitness diploids, Ras/PKA mutants — are roughly equally represented (see Materials and methods). The remaining 232 mutants comprise the test set. This set contains all mutation types represented by only a single mutant, including all TOR/Sch9 (*TOR1, SCH9, KOG1*) and HOG (*SSK2*) pathway representatives, as well as the rest of the recurrent mutants that were not picked for the training set. We include these diverse mutants in the test set so that we can measure the ability of our genotype-phenotype-fitness model to predict the fitness of mutants in genes and pathways that are absent from the training set.

We iteratively construct phenotype spaces using the 60 training mutants while holding out one subtle perturbation at a time and creating the space with the data from the remaining 24 subtle perturbations. We then predict the fitness of the 232 held-out testing mutants in the held-out condition. We do so using all eight components, and again with only 7, 6, and so on. Then, we ask whether the eight component model does a better job at predicting mutant fitness than the other, lower dimensional models. If a component reflects measurement noise rather than biological signal, then the inclusion of this component would lead to overfitting and should harm the model's ability to predict fitness in the held-out data. Instead we find that, on average across the 25 iterations, prediction power improves from the inclusion of each of the eight components. This confirms that even the smallest of these components captures biologically meaningful variation in fitness across the 25 subtle perturbations of the EC. However, the gain in predictive power decreases for each component. The model with only the first component explains on average 85% of weighted variance for the test mutants in the left-out conditions. A model with only the top five components explains 95.1%, and all eight components explain 96.2% of variation. This suggests that the last few components have very small contributions to fitness in the environments near the EC.

## A model including eight fitness-relevant phenotypes recapitulates known features of adaptive mutations

We next ask whether the eight-dimensional phenotypic model clusters adaptive mutants found in similar genes or pathways (e.g. Ras/PKA or TOR/Sch9), or that represent similar mutation types (haploid v. diploid). Alternatively, our model may classify mutations into functional units (i.e. mutations that have similar phenotypic effects) in a way that does not conform to gene or pathway identity. We use Uniform Manifold Approximation and Projection (UMAP) to visualize the distance between all the mutants in this phenotypic space. As the first phenotypic dimension captures the average fitness of each mutant in the EC, and since we already know that mutations to the same gene have similar fitness in the EC (*Figure 2B*), we exclude the first phenotypic dimension from this analysis, although the inclusion of the first component does not change the identity of the clusters (*Figure 3—figure supplement 4A*). By focusing on the other seven components, we are asking whether genotype-by-environment interactions also cluster the mutants by gene, mutation type, and pathway.

These seven genotype-by-environment interactions indeed tend to cluster the adaptive mutants by type and by gene (*Figure 3C*). Specifically, the diploids, *IRA1 nonsense, GPB2,* and *PDE2* mutants each form distinct clusters (p=0.0001, p=0.006, p=0.0001, and p=0.0001, respectively). To generate p-values, we calculated the median pairwise distance, finding that multiple mutations in the same cluster are indeed more closely clustered than randomly chosen groups of mutants. Interestingly, the three smallest components, which capture very little variation in fitness across the environments that reflect subtle perturbations of the EC, also cluster some mutants by gene (*Figure 3—figure supplement 4B*). Specifically, *PDE2, GPB2,* and *IRA1 nonsense* mutants are each closer to mutants of their own type than to other adaptive haploids (p=0.0001, p=0.0001, and p=0.03, respectively). Note that the space defined by the three smallest components does not cluster *IRA1 nonsense* mutants away from diploids (p=0.718). This suggests that some mutants, for example *IRA1 nonsense* and diploids, have smaller effects on these three phenotypic components. Overall, our abstract phenotypic model, which reflects the way that each mutant's fitness changes across environments, reveals that mutations to the same gene tend to interact similarly with the environment.

Our approach also detects cases where mutations to the same gene or pathway do not cluster together. This suggests that our model captures phenotypic effects that would be obscured by assuming mutations to the same gene affect the same traits. For example, genotype-by-environment interactions do not cluster *IRA1 missense* mutations (p=0.317) (*Figure 3C*; light blue points), despite clustering the *IRA1 nonsense* mutations. Perhaps, *IRA1 missense* mutations have more diverse impacts on phenotype than do *IRA1 nonsense* mutations because the latter all likely result in a loss of the IRA1 protein, albeit not necessarily to the same extent. Our model also does not cluster the eight mutations in *IRA2* (p=0.086) (*Figure 3C*; dark gray points). At the pathway level, our model does not cluster the three mutations to the TOR/Sch9 pathway away from the rest of the mutants, which are mainly in the Ras/PKA pathway (p=0.155) (*Figure 3C*; purple points). Our model also does not cluster all diploids that possess additional mutations, including those with increased copy number of chromosome 11 or chromosome 12 and those with mutations in *IRA1* or *IRA2* (p=0.863) (*Figure 3C*; dark red points). Interestingly, our model does find a distinct cluster of six diploids that

have higher than average diploid fitness in the EC (p=0.0001) despite whole genome sequencing having revealed no mutations in their coding sequences (*Figure 3C*). This likely indicates that these diploids harbor difficult-to-sequence additional adaptive mutations that all have similar phenotypic consequences. In sum, these observations suggest that our genotype-phenotype-fitness model reveals new insights about which mutations affect the same functional units, specifically that these units do not always correspond to genes and pathways. Overall, these results suggests that our approach, like others that compare genotype-by-environment interactions (*Li et al., 2018*), is a useful and unbiased way to identify mutations that share functional effects.

## Fitness variation across subtly different environments predicts fitness in substantially different environments

Now that we have identified the phenotypic components that contribute to fitness in environments that represent subtle perturbations of the EC, we can test the ability of these phenotypic components to predict fitness in more distant environments. Specifically, we can measure how the contribution of each of these components to fitness changes in new environments. We can also determine whether the phenotypic components that contribute very little to explaining fitness variation near the EC might at times have large explanatory power in distant environments (as depicted in the 'fitness-relevant modularity' model shown in *Figure 1B*).

To test this we performed bi-cross-validation, using the eight component model constructed from fitness variation of 60 training mutants across 25 subtly different environments to predict the fitness of 232 test mutants in the environments that represent strong perturbations of the EC. To evaluate the predictive power of the model, we compare our model's fitness predictions in each environment to predictions made using the average fitness in that environment. Thus, negative prediction power indicates cases where the model predicts fitness worse than predictions using this average (*Figure 4A*).

The eight-dimensional phenotypic model, which was generated exclusively with the data from subtle environmental perturbations, has substantial predictive power in distant environments (*Figure 4*). Predictions explain 29–95% of the variation in fitness of the 232 test mutants across strong environmental perturbations. For instance, in an environment where glucose concentration was increased from 1.5% to 1.8% and the flask was changed to one that increases the oxygenation of the media (the 'Baffle, 1.8% Glucose' environment), we predict 95% of weighted variance with the full eight-component phenotypic model, in contrast to 51% with a one-component model (*Figure 4B*). This ability to predict fitness is retained even when the first component (effectively the fitness in EC) is a poor predictor of mutant fitness. For example, in the environment where salt (0.5 M NaCl) was added to the media, the one-component model predicts fitness worse than predictions based on the average fitness for this environment, resulting in negative variance explained (*Figure 4A and B*). This is due to the fact that mutant fitness in this environment reflects extensive genotype-by-environment interactions, such that the fitness of mutants in this environment is uncorrelated with EC fitness. However, our predictions of mutant fitness in the 0.5 M NaCl environment improve when made using the eight-component phenotypic model, which predicts 72% of weighted variance. Astoundingly, the eight-component model captures strong tradeoffs between mutants with high fitness in the EC and very low fitness in this high-salt environment, specifically for *IRA1 nonsense* and, to a lesser extent, *PDE2* mutants (*Figure 4B*). This was surprising because there appears to be very little variation in fitness of these mutants across the subtle compared to the strong perturbations (*Figure 2C*).

This ability to predict fitness is also observed for mutations in genes and pathways that are not represented in the 60 that comprise the training set (e.g. those with mutations in TOR/Sch9 and HOG pathway genes). For example, the eight-component model explains 93% of variation in the 'Baffle, 1.8% Glucose' environment and 71% of variation in the 0.5M NaCl environment for these mutations, compared to 76% and 31% variance explained for the one-component model, respectively. This indicates that our model is able to capture shared phenotypic effects that extend beyond gene identity. Altogether, our ability to accurately predict the fitness of new mutants in new environments suggests that the phenotypes our model identifies reflect causal effects on fitness.

Most strikingly, phenotypic models that include the three smallest phenotypic components, which together contribute only 1.1% to variance explained across the subtle environmental perturbations (*Figure 4A*), often explain a substantial amount of variance in the distant environments (*Figure 4A*;

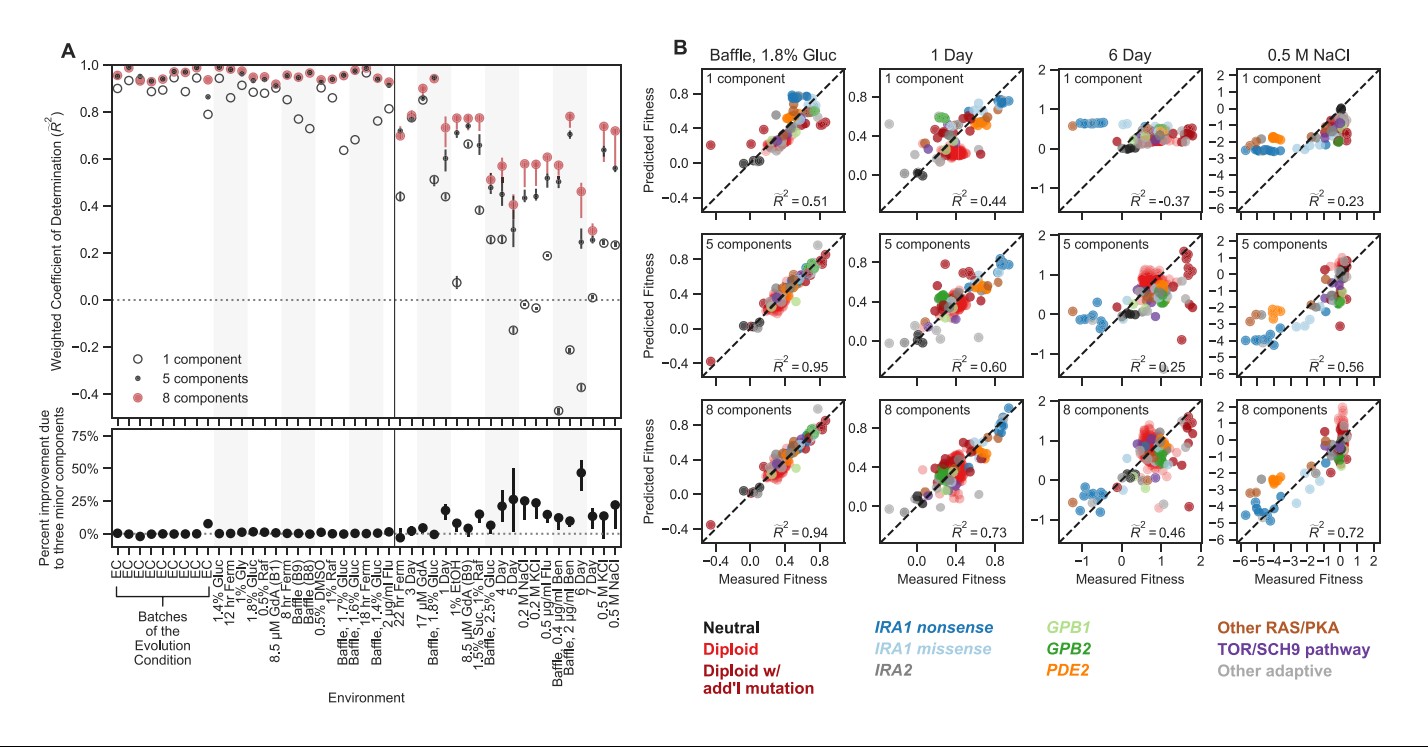

**Figure 4.** Mutant fitness variation across subtly different environments predicts mutant fitness in novel and substantially different environments. (**A**) Top panel vertical axis shows the accuracy of fitness predictions in each of 45 environments on the horizontal axis. The accuracy is calculated as the coefficient of determination, weighted such that each mutation type contributes equally. The left side of this plot represents predictions of mutant fitness in subtle environmental perturbations. These predictions are generated by holding out data from that environment when building the phenotypic model. The right side of the plot displays predictions of mutant fitness in strong environmental perturbations. These predictions are generated using a phenotypic model inferred from fitness variation across all 25 subtle different environments (denoted by each of the points or open circles) and for each of the 25 leave-one-out models (range of predictions is depicted with the error bars surrounding each point or open circle). Predictions from the eight-component model (red point) are typically better than the one-component model (open circle) and sometimes better than the five-component model (black point). **Bottom panel** vertical axis shows the percent of the eight-component model's improvement due to the three minor components (calculated by the percent difference between the five- and eight-component models). The left side shows the improvement of the prediction in subtle environmental perturbations when that subtle perturbation was held out. The right side shows the improvement of the prediction in strong environmental perturbations when using the full model (dots) or the 25 leave-one-out models (the error bars represent the range of improvement). (**B**) For each subplot, the horizontal axis shows the measured fitness value. The vertical axis shows the predicted fitness value when predictions are made using the one-component (top row), five-component (middle row), or eight-component (bottom row) models. Columns represent different environments. Points are colored by the mutation type. Note that $\tilde{R}^2$ less than zero indicates that the prediction is worse than predictions using the mean fitness in that condition (see Materials and methods).

The online version of this article includes the following figure supplement(s) for figure 4:

**Figure supplement 1.** Number of detected components and predictive power increase with the number of training environments.

**Figure supplement 2.** Predictive power increases with the number of mutation types included.

**Figure supplement 3.** Improved fitness predictions when including the three smallest phenotypic components is not specific to choice of training mutants.

**Figure supplement 4.** Prediction ability using unweighted coefficient of determination.

lower panel). For example, the three minor components contribute 17% of the overall weighted variance explained in the 1 Day condition ($\tilde{R}^2$ = 0.6—5-component model, $\tilde{R}^2$ = 0.73—8-component model; (0.73–0.6)/0.73 = 0.17) and 45% in the 6-Day environment, ($\tilde{R}^2$ = 0.25—5-component model, $\tilde{R}^2$ = 0.46—8-component model) (*Figure 4A and B*). In contrast, for other strong environments (e.g. Baffle — 1.8% Glucose, 8.5 µM GdA (B9) and Baffle — 2.5% Glucose), the three smallest components do not add much explanatory power (*Figure 4A*). These observations demonstrate that phenotypic components that make very small contributions to fitness in the EC can contribute substantially to fitness in other environments. Overall, these observations suggest an answer to

questions about how adaptation is possible when mutations have collateral effects on multiple phenotypes: not all of those phenotypes contribute substantially to fitness in the EC (*Figure 1B*).

The strength of our predictions depends on how many subtle environments we used to generate our phenotype model. When we use too few, we robustly detect the largest phenotypic components, but lose power to detect minor components, which can lead to less accurate predictions of fitness in strong environmental perturbations. We show this by randomly subsampling our 25 subtle environments and repeating all of our downstream analyses (*Figure 4—figure supplement 1*). We see a similar pattern when we reduce the number of mutation types used in the training set. Randomly excluding many mutation types from the training set decreases our ability to predict fitness, though the exclusion of any one mutation type from the training set has limited impact on our overall predictive accuracy (*Figure 4—figure supplement 2*).

## Idiosyncratic behavior of some mutants in some environments reveals latent phenotypic complexity

Next, we explore the extent to which the contribution of a phenotypic component to fitness is isolated to a specific environment and/or a specific type of mutation (*Figure 5*). We find that many phenotypic components matter more to fitness in some environments than others. For instance, component two adds on average 36% of the weighted variance in fitness across strong perturbations, despite adding only 7% on average across the subtle environmental perturbations. This contribution is, however, variable, with the second component adding over 90% of variance explained for the two environments with Benomyl and Baffled flasks (the 'Baffle, 0.4 µg/mL Benomyl' and 'Baffle, 2 µg/mL Benomyl' environments) and only 0.3% for the environment in which the transfer time was lengthened from 2 to 3 days (*Figure 5A*).

This environment-dependence is also true for the smallest two components. Specifically, predictions of mutant fitness in the 0.5 M NaCl environment are improved from the inclusion of component 7, adding 7.5% to weighted variance explained (*Figure 5A*). Predictions of mutant fitness in the 6-Day transfer environment show improvement from the inclusion of the 8th component, which adds over 15% to weighted variance explained (*Figure 5A*). However, the predictions of fitness in the 6 Day environment are not improved from the inclusion of the 7th component and the predictions in 0.5 M NaCl are not improved markedly by the inclusion of the 8th component (*Figure 5A*). This suggests that the phenotypic effects represented by these small components contribute substantially in some environments and not others.

We further asked whether these effects are not only environment-specific but also mutant-specific. To do so, we focused on environments for which the two smallest components contribute substantially to fitness (e.g. 0.5 M NaCl). We looked at the extent to which each of these components improves power to predict the fitness of each of the 232 held-out mutants. We found these components improve the fitness predictions for some classes of mutants far more than for others. For example, fitness predictions for mutations in *GPB2*, diploids with chromosome 11 amplifications, and high-fitness diploids with no known mutations each improved by over four standard deviations of measurement error in the 0.5 M NaCl environment due to the inclusion of the 7th component (*Figure 5B*). This phenotypic component also has importance in the 1 Day transfer environment, albeit to a lesser degree, resulting in improvements of roughly one standard deviation for each of these mutation types. This suggests that these mutants have some phenotypic effect that contributes only slightly to fitness in many environments, including those that represent subtle perturbations of the EC, but that are particularly important in the 0.5 M NaCl and 1-Day transfer environments. Similarly, we find that the 8th component also improves predictive power for specific types of mutants in specific environments. In this case, diploids with chromosome 11 amplifications and *PDE2* mutants have particularly strong improvements in the 6-Day transfer environment (11 and 5 standard deviations, respectively) and thus likely have a shared phenotypic effect that is captured by component 8 (*Figure 5B*).

In sum, not all mutations affect all eight phenotypic components to the same degree and not all phenotypic components contribute substantially to fitness in all environments. This idiosyncrasy suggests that directional selection has the potential to generate rather than reduce phenotypic diversity in cases where multiple adaptive mutants persist within a population or across populations. Although directional selection 'chooses' mutations that affect a small number of similar phenotypes relevant to fitness in the EC, these mutations may have latent effects on a larger number of diverse

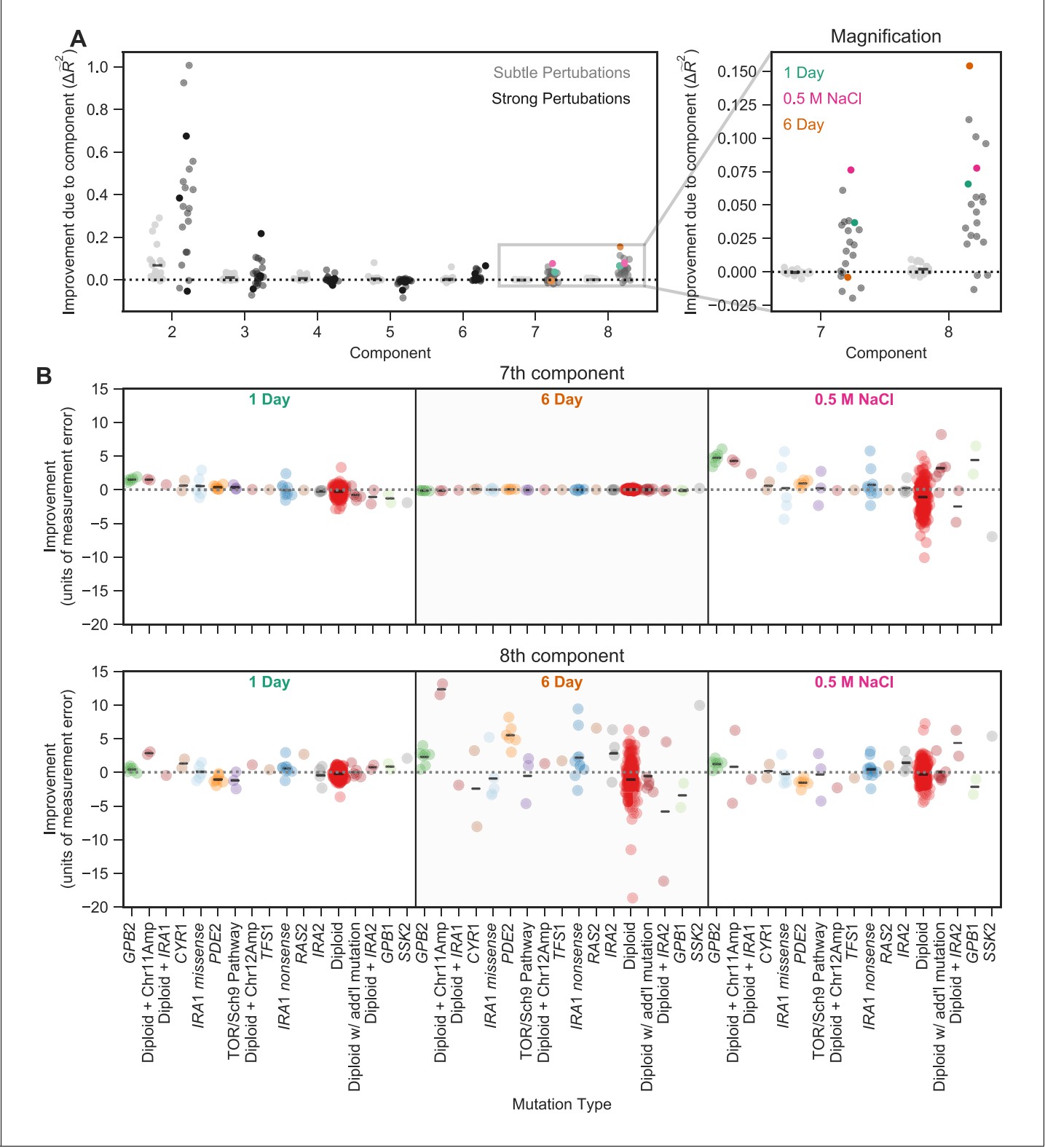

**Figure 5.** The contribution of a phenotypic component to fitness changes across environments and differs for different types of mutants. (**A**) Some phenotypic components improve fitness predictions in some environments substantially more than they do in others. The vertical axis shows the improvement in the predictive power of our eight-component phenotypic model due to the inclusion of each component. For example, the improvement due to component seven is calculated by the difference between the seven-component model and the six-component model. The improvement of predictive power for each of the subtle environmental perturbations is shown as a gray point and for each of the strong perturbations

*Figure 5 continued on next page*

*Figure 5 continued*

in black. Magnification shows improvement upon including each of the two smallest components, with three strong perturbations highlighted. (**B**) Some phenotypic components improve fitness predictions for some mutants substantially more than they do for others. For example, the 7th component explains little variation in the 6-Day environment, but the 8th component explains a lot of variation in fitness in the 6-Day environment and is particularly helpful in predicting the fitness of Diploid + Chromosome 11 Amplification mutations in this environment. Vertical axis shows the improvement in predictive power (in units of standard deviation of measurement error) for each type of mutant (denoted on the horizontal axis) in one of three environments (1 Day, 6 Day, and 0.5 M NaCl) when adding either the 7th (top panel) or the 8th (bottom panel) component. Mutants are ordered by the improvement due to the 7th component in the 1-Day environment. Since some types of mutants are more common, for example diploids, there are more data points in that category.

phenotypes. When the environment changes, these latent phenotypic effects are revealed, exposing the phenotypic diversity generated by the adaptive process.

## Discussion

Here, we succeeded in building a low-dimensional statistical model that captures the relationship from genotype to phenotype to fitness for hundreds of adaptive mutants. Mapping the complete phenotypic and fitness impacts of genetic change is a key goal of biology. Such a map is important in order to make meaningful predictions from genetic data (e.g. personalized medicine) and to investigate the structure of biological systems (e.g. their degree of modularity and pleiotropy) (*Collet et al., 2018*; *Eguchi et al., 2019*; *Exposito-Alonso et al., 2019*; *Zan and Carlborg, 2020*). Our model allows us to do both of these things. We made accurate predictions about the fitness of unstudied mutants across multiple environments, and we gained novel insights about the degree to which adaptive mutations are modular versus pleiotropic. Specifically, we learned that adaptation is modular in the sense that hundreds of diverse adaptive mutants collectively influence a small number of phenotypes that matter to fitness in the evolution condition. We also learned that different mutants have distinct pleiotropic side effects that matter to fitness in other conditions.

Building genotype-phenotype-fitness maps of adaptation has long been an elusive goal due to both conceptual and technical difficulties. Indeed, the very first part of this task, namely the identification of causal adaptive mutations, presents a substantial technical challenge (*Barrett et al., 2019*; *Barrett et al., 2008*; *Exposito-Alonso et al., 2019*). Fortunately, in some systems, such as in microbial experimental evolution and studies of cancer and resistance in microbes and viruses, genomic methodologies combined with availability of repeated evolutionary trials allow us to detect specific genetic changes responsible for adaptation. In the context of microbial evolution experiments, lineage tracing and genomics have opened up the possibility of not only detecting hundreds of specific adaptive events but also measuring their fitness precisely and in bulk (*Good et al., 2017*; *Levy et al., 2015*; *Li et al., 2019*; *Li et al., 2018*; *Nguyen Ba et al., 2019*; *Venkataram et al., 2016a*). Thus, in these cases, we are coming close to solving the technical challenge of building the *genotype to fitness* map of adaptation.

However, adding *phenotype* into this map remains a huge challenge even despite substantial progress in mapping genotype to phenotype (*Burga et al., 2019*; *Camp et al., 2019*; *Exposito-Alonso et al., 2018*; *Geiler-Samerotte et al., 2016*; *Jakobson and Jarosz, 2019*; *Lee et al., 2019*; *Paaby et al., 2015*; *Yengo et al., 2018*; *Ziv et al., 2017*). In principle, we now have advanced tools to measure a large number of phenotypic impacts of a genetic change, for instance through high-throughput microscopy, proteomics, or RNAseq (*Manzoni et al., 2018*; *Ritchie et al., 2015*; *Zhang and Kuster, 2019*). The conceptual problem is how to define phenotypes given the interconnectedness of biological systems (*Geiler-Samerotte et al., 2020*; *Paaby and Rockman, 2013*). If a mutation leads to complex changes in cell size and shape, should each change be considered a distinct phenotype? Or if a single mutation changes the expression of hundreds or thousands of genes, should we consider each change as a separate phenotype? Intuitively, it seems that we should seek higher order, more meaningful descriptions. For example, perhaps these expression changes are coordinated and reflect the upregulation of a stress-response pathway. Unfortunately, defining the functional units in which a gene product participates remains difficult, especially because these units re-wire across genetic backgrounds, environments, and species (*Geiler-Samerotte et al., 2020*; *Pavličev et al., 2017*; *Sun et al., 2020*; *Zan and Carlborg, 2020*).

If mutations influence more than one phenotype, then the mapping from phenotype-to-fitness also becomes challenging. To investigate this map, we would need to find an artificial way to perturb one phenotype without perturbing others such that we could isolate and measure effects on fitness. Mapping phenotype to fitness is further complicated by the environmental dependence of these relationships (*Fragata et al., 2019*; *Price et al., 2018*). For example, a mutation that affects a cell's ability to store carbohydrates for future use might matter far more in an environment where glucose is re-supplied every 6 days instead of every 48 hr.

In our study, we turned the challenge of environment-dependence into the solution to the seemingly intractable problem of interrogating the phenotype layer of the genotype-phenotype-fitness map. We rely on the observation that the relative fitness of different mutations changes across environments. We assume that differences in how mutant fitness varies across environments must stem from differences in the phenotypes each mutation affects. Rather than a priori defining the phenotypes that we think may matter, we use the similarities and dissimilarities in the way fitness of multiple mutants vary across environments to define phenotypes abstractly via their causal effects on fitness. This allows us to dispense with measuring the phenotypes themselves and instead focus on measuring fitness with high precision and throughput, since tools for doing so already exist (*Venkataram et al., 2016a*). This approach has the disadvantage of not identifying phenotypes in a traditional, more transparent way. Still, it represents a major step forward in building genotype-phenotype-fitness maps because it makes accurate predictions and provides novel insights about the phenotypic structure of the adaptive response.

We successfully implemented this approach using a large collection of adaptive mutants evolved in a glucose-limited condition. The first key result is that the map from adaptive mutant to phenotype to fitness is modular, such that it is possible to create a genotype to phenotype to fitness model that is low dimensional. Indeed, our model detects a small number (8) of fitness-relevant phenotypes, the first two of which explain almost all of the variation in fitness (98.3%) across 60 adaptive mutants in 25 environments representing subtle perturbations of the glucose-limited evolution condition. This suggests that the hundreds of adaptive mutations we study — including mutations in multiple genes in the Ras/PKA and TOR/Sch9 pathways, genome duplication (diploidy), and various structural mutations — influence a small number of phenotypes that matter to fitness in the evolution condition. This observation is consistent with theoretical considerations suggesting that mutations that affect a large number of fitness-relevant phenotypes are not likely to be adaptive (*Orr, 2000*; *Wagner and Altenberg, 1996*). It also explains findings from other high-replicate laboratory evolution experiments and studies of cancer that show hundreds of unique adaptive mutations tend to hit the same genes and pathways repeatedly (*Hanahan and Weinberg, 2011*; *Hanahan and Weinberg, 2000*; *Sanchez-Vega et al., 2018*; *Tenaillon et al., 2012*; *Venkataram et al., 2016a*). Our work confirms the intuition that these mutations all affect similar higher-order phenotypes (e.g. the level of activity of a signalling pathway). This suggests that, despite the genetic diversity among adaptive mutants, adaptation may be predictable and repeatable at the phenotypic level.

Note that although we detect only eight fitness-relevant phenotypes, we expect the true number to be much larger as the detectable number is limited by the precision of measurement (see Materials and methods and *Figure 2—figure supplement 1*) and the number of environments used to construct the phenotypic model (*Figure 4—figure supplement 1*). We expect this partly because we know that if we had worse precision in this experiment we would have detected fewer than eight phenotypic components (*Figure 3*). Still, these additional undetected components cannot be very consequential in terms of their contribution to fitness in the evolution condition, given how well the first eight components capture variation in environments that are similar to the evolution condition.

Surprisingly, the model built only using subtle environmental perturbations was also predictive of fitness in environments that perturbed fitness strongly. In some of these environments, such as the environment where 0.5 M NaCl was added to the media or the time of transfer was extended from 2 to 6 days, many of the mutants are no longer adaptive and some of them become strongly deleterious. Here, the fitness of the mutants in the evolution condition is a very poor predictor of fitness. Despite this, the eight-dimensional phenotypic model built using subtle perturbations of the evolution condition explains from 29% to 95% of the variance in environments that represent strong perturbations. What was particularly interesting is that the explanatory power of different phenotypic components was very different for the strong compared to subtle perturbations. For instance, the second component, which explained 7% of weighted variation on average in the subtle

perturbations, explained 36% on average in the environments that represent strong perturbations. The pattern was particularly striking for the smallest three components which at times explained 15% in the strong environmental perturbations while again explaining at most 1% in the subtle environmental perturbations.

This discovery emphasizes that, although the smaller phenotypic components contribute very little to fitness in the evolution condition, they can at times have a much larger contribution in other environments, as predicted by the fitness-relevant modularity model (*Figure 1B*). This makes intuitive sense. For instance, we know that some of the strongest adaptive mutations in our experiment, the nonsense mutations in IRA1, appear to stop cells from shifting their metabolism toward carbohydrate storage when glucose levels become low (*Li et al., 2018*). This gives these cells a head start once glucose again becomes abundant and does not appear to come at a substantial cost, at least not until these cells are exposed to stressful environments (e.g. high salt or long stationary phase) (*Li et al., 2018*). This example, and more generally the observation that phenotypic effects that are unimportant in the evolution condition can become more important in other environments, supports the idea that adaptation can happen through large effect mutations because many of the pleiotropic effects will be inconsequential in the local environment (*Figure 1B*). We can thus argue that our low-dimensional model representing the genotype-phenotype-fitness map near the evolution condition hides consequential phenotypic complexity across the collection of adaptive mutants. This complexity is hidden from natural selection in the evolution condition but becomes important once the mutants leave the local environment and are assessed globally for fitness effects. Thus, with respect to their effects on fitness-relevant phenotypes, adaptive mutants may be locally modular, but globally pleiotropic.

The notion of latent phenotypic complexity is exciting as it generates a mechanism by which directional selection generates rather than removes phenotypic diversity. Although directional selection may promote multiple mutants that affect similar fitness-relevant phenotypes in the evolution condition, each mutant could have disparate latent phenotypic effects that do not contribute immediately to fitness. When the environment changes, these disparate phenotypic effects may be revealed, imposing fitness costs of different magnitudes or allowing for diverse solutions to a variety of possible new environments (*Bono et al., 2017*; *Chavhan et al., 2020*; *Jerison et al., 2020*; *Li et al., 2019*). This latent phenotypic complexity also has the potential to alter the future adaptive paths that a population takes even in a constant environment. Indeed, these phenotypically diverse mutants are likely to affect the subsequent direction of adaptation given that subsequent mutations can shift the context in which phenotypes are important in the same way as do environmental perturbations (*Blount et al., 2018*; *Blount et al., 2008*; *Dillon et al., 2016*). Latent phenotypic complexity among adaptive mutations is thus similar to cryptic genetic variation in that it can influence a population's ability to adapt to new conditions (*Paaby and Rockman, 2013*), but dissimilar in that it evolves under directional rather than stabilizing selection. The end result is that directional selection can generate diversity both within a population in which multiple adaptive mutants are segregating and across populations that are adapting to the same stressors.

The phenomenon of latent phenotypic complexity being driven by adaptation is dependent on there being multiple mutational solutions to an environmental challenge, such that different adaptive mutations might have different latent phenotypic effects. Latent phenotypic diversity might be less apparent in cases where adaptation proceeds through mutations in a single gene and certainly would not exist if adaptation relies on one unique mutation. Thus, in some ways, latent phenotypic diversity reflects redundancies in the mechanisms that allow cells to adapt to a challenge. One such putative redundancy in the case investigated in this paper is that the Ras/PKA pathway can be constitutively activated by loss-of-function mutations to a number of negative regulators including IRA1, PDE2, and GPB2. Mutations in these genes might be redundant in the sense that they influence the same fitness-relevant phenotype in the evolution condition, which in this case is likely flux through the Ras/PKA pathway. This type of redundancy is commonly observed in laboratory evolutions (*Barghi et al., 2020*) and is particularly apparent in studies that analyze individuals with several adaptive mutations. Such studies find that multiple mutations in the same functional unit occur less than expected by chance presumably because those mutations would have redundant effects on fitness (*Tenaillon et al., 2012*). Similarly, studies also find that second-step adaptive mutations tend to be in different pathways or functional modules than the first adaptive step (*Aggeli et al., 2020*; *Fumasoni and Murray, 2020*). The novel observation from our paper is that mutations with

redundant effects on fitness in the evolution condition are not necessarily identical because they may influence different latent phenotypes. This observation adds to a long list of examples demonstrating that redundancies, such as gene duplications and dominance, allow evolution the flexibility to generate diversity.

One disadvantage of our approach is that the phenotypic components that we infer from our fitness measurements are abstract. They represent causal effects on fitness, rather than measurable features of cells. For this reason, perhaps we should not refer to them as phenotypes but rather 'fitnotypes' (a mash of the terms 'fitness' and 'phenotype') that act much like the causal traits in Fisher's geometric model (*Blanquart et al., 2014*; *Blanquart and Bataillon, 2016*; *Fisher, 1930*; *Harmand et al., 2017*; *Lourenço et al., 2011*; *Martin and Lenormand, 2006*; *Poon and Otto, 2000*; *Tenaillon, 2014*; *Tenaillon et al., 2007*; *Weinreich and Knies, 2013*) or a selectional pleiotropy model (*Paaby and Rockman, 2013*). Despite this limitation, these fitnotypes have proven useful in allowing us to understand the consequences of adaptive mutation. In addition to insights discussed above, we also learned that adaptive mutants in the same gene do not always affect the same fitnotypes. For example, we found that *IRA1 missense* mutations have varied and distinct effects from *IRA1 nonsense* mutations. Another way that identifying fitnotypes may ultimately prove useful is in identifying the phenotypic effects of mutation. The fitnotypes can serve as a scaffold onto which a large number of phenotypic measurements can be mapped. Even though fitnotypes are independent with respect to their contribution of fitness and contribute to fitness linearly, the mapping of commonly measured features of cells (e.g. growth rate, the expression levels of growth supporting proteins like ribosomes) onto fitnotypes may not be entirely straightforward. Nonetheless, methods such as Sparse Canonical Correlation Analysis (*Suo et al., 2017*) hold promise in such a mapping and might help us relate traditional phenotypes to fitnotypes.

An important question for future research is whether our observation of local modularity and global pleiotropy are also apparent in other cases of adaptation. The method we described is generic and can be applied to any system as long as the fitness of a substantial set of mutants can be profiled across multiple environments or genetic backgrounds. This is becoming possible to do in many systems (*Flynn et al., 2020*; *Jerison et al., 2020*; *Li et al., 2019*; *Martin et al., 2015*; *Pan et al., 2018*; *Rogers et al., 2018*) and presents an opportunity to understand how the number of fitness-relevant phenotypes that a collection of mutations affects depends on the environment in which those mutations evolved and the environment in which their fitness effects are assessed.

The notion that diverse genetic changes can have redundant effects in one environment but distinct and consequential effects in other environments is important to our understanding of adaptation in other settings, including in the context of antibiotic resistance and cancer. For example, tumors representing the same type of cancer (e.g. lung adenocarcinoma) tend to be genetically diverse even if considering only driver mutations (*Cancer Genome Atlas Research Network, 2014*). However, the driver mutations often fall into a smaller number of key driver genes and even fewer pathways (*Bailey et al., 2018*; *Hanahan and Weinberg, 2011*; *Hanahan and Weinberg, 2000*; *Sanchez-Vega et al., 2018*; *Sondka et al., 2018*). While this apparent redundancy might suggest that the tumors are functionally similar, the notion of latent diversity we propose here suggests that the specific mutational paths taken by different tumors might matter once the environment changes, for example when the tumors are treated by a cancer therapy. Substantial heterogeneity of tumor response to therapy is consistent with this notion (*Li et al., 2020*).

Despite the accumulation of large amounts of genomic and phenomic data, integrating this information to identify the phenotypic consequences of mutation that are ultimately responsible for fitness remains incredibly challenging. Our approach allows us to create an abstract representation of the causal effects of genetic mutation and their changing contribution to fitness across environments. This top-down view of the genotype-phenotype-fitness map simplifies the complex and multifaceted phenotypic consequences of mutation by focusing on those that contribute to fitness. Integrating this new perspective with the influx of precise and high-throughput data might allow us to answer age-old questions about the structure of biological systems and adaptation.

# Materials and methods

## Key resources table

| Reagent type (species) or resource | Designation | Source or reference | Identifiers | Additional information |
|---|---|---|---|---|
| Commercial assay or kit | OneTaq Hot Start 2X Master Mix with Standard Buffer | New England Biolabs | Cat#M0484L | |
| Commercial assay or kit | Q5 DNA Polymerase | New England Biolabs | Cat#M0491L | |
| Commercial assay or kit | ApaLI restriction enzyme | New England Biolabs | Cat#R0507L | |
| Commercial assay or kit | MasterPure Yeast DNA Purification Kit | Lucigen | Cat#MPY80200 | |
| Strain, strain background (*Saccharomyces cerevisiae*) | *S. cerevisiae* constructed reference strain | *Venkataram et al., 2016a* | GSY 6704 | |
| Commercial assay or kit | Nextera XT Index Kit v2 | Illumina | Cat#FC-131–2004 | |
| Sequence-based reagent | Primers F201-F212 and R301-R308 | This paper | Step 1 PCR primers | See Materials and methods section 'PCR Amplification of the Barcode Locus' |
| Software, algorithm | Pipeline to determine the number of barcode reads | *Venkataram et al., 2016a* | | |
| Software, algorithm | Pipeline to calculate fitness from barcode counts | *Venkataram et al., 2016a* | | |

## Lead contact and materials availability

Further information and requests for resources and reagents should be directed to and will be fulfilled by the Lead Contact, Dmitri Petrov (dpetrov@stanford.edu).

## Experimental model and subject details

The yeast strains used in this study can be grown and maintained using standard methods (e.g. YPD media in test tubes, glycerol stocks for long term storage at −80°C), but should be propagated in the appropriate selection environment (a glucose-limited minimal media - M3 medium for the evolution condition) for comparable fitness and phenotypic measurements. All the strains we study are of genetic background MATα, ura3Δ0, ybr209w::Gal-Cre-KanMX-1/2URA3-loxP-Barcode-1/2URA3-HygMX-lox66/71.

Experiments were performed with barcoded mutants isolated from a previous evolution experiment (*Levy et al., 2015*). To measure their fitness, these mutants were competed against a constructed reference strain with a restriction site in the barcode region (*Venkataram et al., 2016a*).

The majority of the fitness measurement experiments were conducted with a collection of 500 adaptive barcoded mutants where each strain starts at equal frequency (*Li et al., 2018*; *Venkataram et al., 2016a*). We focus on a subset of 292 strains for which we obtained fitness measurements in all 45 environments and for which mutations conferring fitness advantages have been previously identified, either by whole genome sequencing or using a drug to test ploidy (*Li et al., 2018*; *Venkataram et al., 2016a*; *Supplementary file 1*). Note that because we utilize some data from previous experiments (*Li et al., 2018*; *Venkataram et al., 2016a*), some of the experiments contained additional barcoded mutants not analyzed here, namely a pool consisting of a total of 4800 strains, including the 292 focused on in this study. These differences in the number of strains included in the experiment are partially accounted for in our inference of mean fitness, and any remaining effects can be thought of as another parameter that varies across the environments (e.g. in addition to glucose or salt concentration).

In a few experiments, we spiked in re-barcoded mutants and additional neutral lineages as internal controls. Since re-barcoded mutants are identical, except for the barcode, these teach us about the precision with which we can measure a mutant's fitness. Specifically, we spiked in 10 re-barcoded *IRA1 nonsense* mutants (each with a frameshift insertion AT to ATT mutation at bp 4090) and 10 *IRA1 missense* mutants (each with a G to T mutation at bp 3776). Neutral lineages teach us about the behavior of the unmutated reference strain, which we must infer because its barcode is eliminated from the experiment before sequencing. The spiked in neutrals include ten barcoded lineages

from the original evolution experiment (*Levy et al., 2015*) for which whole genome sequencing did not reveal any mutations (*Venkataram et al., 2016a*) and previous fitness measurements did not reveal any deviation from the reference (*Li et al., 2018*; *Venkataram et al., 2016a*).

## Method details

### Conducting the barcoded fitness measurements

Fitness measurement experiments were performed as described previously (*Li et al., 2018*; *Venkataram et al., 2016a*), where growth competitions were set up between a pool of barcoded mutants and a reference strain. The change in the frequency of each barcode over time reflects the fitness of the adaptive mutant possessing that barcode, relative to the reference strain.

We conducted fitness measurements under a variety of conditions (*Supplementary file 2*) that represent perturbations of the condition in which these adaptive mutants evolved. Briefly, we separately grew up an overnight culture of the barcode pool and the ancestral reference strain in 100 mL M3 (minimal, glucose-limited) medium (*Verduyn et al., 1992*). We then mixed these saturated cultures at a 1:9 ratio such that 90% of cells represent the reference strain. This ratio allows for mutants to compete against the ancestor rather than competing against each other, helps to minimize the change in average fitness throughout the competition experiment, and reduces the effect of any frequency-dependent fitness effects as well as any fitness-affecting interactions among the strains that may occur. We then inoculated 400 μL of this mixed culture (~5 × 10$^7$cells) into 100 mL of fresh media in 500 mL DeLong flasks. The type of media used, and sometimes the shape of the flask, varied depending on condition (*Supplementary file 2*). This culture was then grown at 30°C in an incubator shaking at 223 RPM for 48 hr. After 48 hr of growth, 400 μL of saturated culture was transferred into fresh media of the same type, in a new flask of the same type. This serial dilution was usually continued four times, yielding five time-points over which to measure the rate at which a barcode's frequency changed, although some experiments include one more or one less depending on the experimenter and on whether technical problems (e.g. PCR failure) caused loss of time-points.

After each transfer of 400 μL, the left-over 9600 μL was frozen so that we could later sequence the barcodes present at every time-point. To prepare this culture for freezing, it was transferred to 50 mL conicals, spun down at 3000 rpm for 5 min, resuspended in 5 mL of sorbitol freezing solution (0.9 M sorbitol, 0.1 M Tris-HCL pH 7.5, 0.1 M EDTA pH 8.0), aliquoted into three 1.5 mL tubes, and stored at −80°C.

For experiments where additional neutral lineages and re-barcoded lineages were included, the initial inoculation mix consisted of 90% ancestral reference strain, 9.4% barcode mutant pool, 0.2% additional neutral spike-in pool, 0.2% re-barcoded IRA1 nonsense pool, and 0.2% re-barcoded IRA1 missense pool.

### Growth conditions

In this study, we present fitness measurement data from a collection of 45 conditions that each represent perturbations of the growth condition in which these adaptive mutants evolved. We refer to this original evolution condition as the 'EC'. In the EC, cells are grown in flasks with a flat bottom and transferred to new flasks every 48 hr (see *Conducting the barcoded fitness measurements*). Cells are grown in M3 media (*Verduyn et al., 1992*). This media is glucose-limited, meaning the cells run out of glucose before any other nutrient. In the EC, the starting glucose concentration is 1.5%.

The 45 perturbations of the EC are summarized in *Supplementary file 2* and include changes to the growth media, the flask shape, and the transfer times. For example, in the '1 Day' condition, we change the transfer time from 48 to 24 hr. In the '1.8% glucose, baffled flask condition' we change the starting glucose concentration from 1.5% to 1.8% and change the flask type from one with a flat bottom to one with baffles. Several of these conditions include experiments from previous studies (*Li et al., 2018*; *Venkataram et al., 2016a*).

For each of these 45 conditions but three, we include between two and four replicates that were performed simultaneously (*Supplementary file 2*) such that overall we performed a total of 109 fitness measurements on our collection of adaptive mutants. Our replicate structure is nested in that some of our 45 conditions represent replicate experiments that we performed at different times. Variation across experiments performed at different times is often referred to as 'batch effects' and

likely reflects environmental variability that we were unable to control (e.g. slight fluctuations in incubation temperature due to limits on the precision of the instrument). In particular, we re-measured the fitness of the adaptive mutants in the EC on nine different occasions, each time including three or more replicates. We refer to these nine experiments as 'EC batches' in the main text. However, every set of experiments that was performed at the same time constitutes a separate 'batch'. There were slight differences across batches in the way we prepared barcodes for sequencing, which we detail in the relevant Methods sections. This variation across batches can be thought of as another parameter that varies across the 45 conditions (in addition to glucose or salt concentration). We report which experiments were performed in the same batch in *Supplementary file 2*.

Some conditions, including some Fluconazole conditions and Geldanamycin conditions, have unexpected orderings in the strength of perturbation (i.e. the smaller drug concentration shows a larger difference in fitness or similar concentrations seem to have different effects). Regardless of whether these observations reflect technical problems (e.g. degradation or poor solubility of the drug), we include these conditions because we use the effect of the realized perturbation on fitness to build low-dimensional phenotypic models. In other words, the identity of the perturbation does not matter in this study.

## DNA extraction of each sample

After a growth competition is complete, we extracted DNA from frozen samples following either a protocol described previously (for batches 1–6 and 10) (*Venkataram et al., 2016a*) or a modified protocol that improves the ease and yield of extraction. Our modified protocol is as follows. For each sample, a single tube of the three that were frozen for each sample (see *Conducting the barcoded fitness measurements*) was removed from the freezer and thawed at room temperature. We extracted DNA from that sample using the following modification of the Lucigen MasterPure yeast DNA purification kit (#MPY80200). We transferred the thawed cells into a 15 mL conical and centrifuge for 3 min at 4000 RPM. After discarding the supernatant, the pellet was then resuspended with 1.8 mL of the MasterPure lysis buffer, and 0.5 mm glass beads were added to help with disruption of the yeast cell wall. The mix of pellet, lysis buffer, and beads was then vortexed for 10 s and incubated for 45 min at 65°C, with periodic vortexing. The solution was then put on ice for 5 min and then 900 µL of MPC Protein Reagent was mixed with the solution. We then separated protein and cell debris by centrifugation at 4000 RPM, transferring 1900 µL of supernatant to a 2 mL centrifuge tube. We further separated remaining protein and cell debris by centrifuging at 13,200 RPM for 5 min. The supernatant was then divided into two 2 mL centrifuge tubes, with 925 µL of the supernatant into each. Next, we added 1000 µL of isopropanol to each tube, mixed by inversion, centrifuged at 13,200 RPM for 5 min, and discarded the supernatant. The pellet, containing the DNA was then resuspended in 250 µL of Elution Buffer and 10 µL of 5 ng/µL RNAase A was added. This was either left at room temperature overnight or incubated at 60°C for 15 min. Next the two tubes per sample were combined into a single tube and 1500 µL of ethanol was added. This was then mixed by inversion, and strands of precipitating DNA appeared. This was centrifuged at 13200 RPM for 2 min, and the supernatant was discarded. We again precipitated the DNA by resuspending with 750 µL of ethanol, and collected the DNA by centrifuging 13200 RPM for 2 min. The supernatant was discarded, and the tubes were left to air dry. Finally, we resuspended the pellet in Elution Buffer to a final concentration of 50 ng/µL for later use in PCR reactions (approximately 3600 ng of DNA were used for the PCR reactions).

## PCR amplification of the barcode locus

After extracting DNA, we PCR-amplified the barcode locus for each sample. Batches 1–6 and 10 were conducted with the protocols described in *Li et al., 2018*; *Venkataram et al., 2016a*. We made some slight modifications to this protocol, including using a new set of primers to allow for nested-unique-dual index labeling, for batches 7, 8, and 9. Our modified protocol is as follows.

We used a two-step PCR protocol to amplify the barcodes from the DNA. The first PCR cycle uses primers with 'inline indices' to label samples (see *Mitigating the effects of index hopping* section for details). These inline indices are highlighted in bold below. Attaching unique indices to samples pertaining to different conditions or timepoints allows us to multiplex these samples on the same sequencing lane. Each primer also contains a Unique Molecular Identifier (UMI) – denoted by

the sequence of 'N' nucleotides in the primer – which is used to determine if identical barcode sequences each represent yeast cells that were present at the time the sample was frozen, or a PCR amplification of the a barcode from a single cell (see *Levy et al., 2015*; *Li et al., 2018*; *Venkataram et al., 2016a*). Primers were HPLC purified to ensure they are the correct length.

## Forward primers

| Primer name | Sequence |
| --- | --- |
| F201 | TCGTCGGCAGCGTC AGATGTGTATAAGAGACAG NNNNNNNN **CGATGTT** TAATATGGACTAAAGGAGGCTTTT |
| F202 | TCGTCGGCAGCGTC AGATGTGTATAAGAGACAG NNNNNNNN **ACAGTGT** TAATATGGACTAAAGGAGGCTTTT |
| F203 | TCGTCGGCAGCGTC AGATGTGTATAAGAGACAG NNNNNNNN **TGACCAT** TAATATGGACTAAAGGAGGCTTTT |
| F204 | TCGTCGGCAGCGTC AGATGTGTATAAGAGACAG NNNNNNNN **GCCAATT** TAATATGGACTAAAGGAGGCTTTT |
| F205 | TCGTCGGCAGCGTC AGATGTGTATAAGAGACAG NNNNNNNN **ATCACGT** TAATATGGACTAAAGGAGGCTTTT |
| F206 | TCGTCGGCAGCGTC AGATGTGTATAAGAGACAG NNNNNNNN **CAGATCT** TAATATGGACTAAAGGAGGCTTTT |
| F207 | TCGTCGGCAGCGTC AGATGTGTATAAGAGACAG NNNNNNNN **GGCTACT** TAATATGGACTAAAGGAGGCTTTT |
| F208 | TCGTCGGCAGCGTC AGATGTGTATAAGAGACAG NNNNNNNN **TAGCTTT** TAATATGGACTAAAGGAGGCTTTT |
| F209 | TCGTCGGCAGCGTC AGATGTGTATAAGAGACAG NNNNNNNN **TTAGGCT** TAATATGGACTAAAGGAGGCTTTT |
| F210 | TCGTCGGCAGCGTC AGATGTGTATAAGAGACAG NNNNNNNN **ACTTGAT** TAATATGGACTAAAGGAGGCTTTT |
| F211 | TCGTCGGCAGCGTC AGATGTGTATAAGAGACAG NNNNNNNN **GATCAGT** TAATATGGACTAAAGGAGGCTTTT |
| F212 | TCGTCGGCAGCGTC AGATGTGTATAAGAGACAG NNNNNNNN **CTTGTAT** TAATATGGACTAAAGGAGGCTTTT |

## Reverse primers

| Primer name | Sequence |
| --- | --- |
| R301 | GTCTCGTGGGCTCGG AGATGTGTATAAGAGACAG NNNNNNNN **TATATACGC** TCGAATTCAAGCTTAGATCTGATA |
| R302 | GTCTCGTGGGCTCGG AGATGTGTATAAGAGACAG NNNNNNNN **CGCTCTATC** TCGAATTCAAGCTTAGATCTGATA |
| R303 | GTCTCGTGGGCTCGG AGATGTGTATAAGAGACAG NNNNNNNN **GAGACGTCT** TCGAATTCAAGCTTAGATCTGATA |
| R304 | GTCTCGTGGGCTCGG AGATGTGTATAAGAGACAG NNNNNNNN **ATACTGCGT** TCGAATTCAAGCTTAGATCTGATA |
| R305 | GTCTCGTGGGCTCGG AGATGTGTATAAGAGACAG NNNNNNNN **ACTAGCAGA** TCGAATTCAAGCTTAGATCTGATA |
| R306 | GTCTCGTGGGCTCGG AGATGTGTATAAGAGACAG NNNNNNNN **TGAGCTAGC** TCGAATTCAAGCTTAGATCTGATA |
| R307 | GTCTCGTGGGCTCGG AGATGTGTATAAGAGACAG NNNNNNNN **CTGCTACTC** TCGAATTCAAGCTTAGATCTGATA |
| R308 | GTCTCGTGGGCTCGG AGATGTGTATAAGAGACAG NNNNNNNN **GCGTACGCA** TCGAATTCAAGCTTAGATCTGATA |

For the first step of PCR, we performed eight reactions per sample to offset the effects of PCR jackpotting within each reaction. For each set of eight reactions, we used the master mix:

- 200 µL OneTaq Hot Start 2X Master Mix with Standard Buffer (NEB M0484L)
- 8 µL 10 uM Forward primer
- 8 µL 10 uM Reverse primer
- 72 µL sample genomic DNA (diluted to 50 ng/µL or all of sample if between 25 and50 ng/µL)
- 16 µL 50 mM MgCl2
- 96 µL Nuclease Free Water (Fisher Scientific #AM9937)

We then aliquoted 50 µL of the master mix into each of eight PCR tubes, and ran on the thermocycler with the following cycle:

1. 94°C for 10 min
2. 94°C for 3 min
3. 55°C for 1 min
4. 68°C for 1 min
5. Repeat steps 2–4 2x (for a total of 3 cycles)
6. 68°C for 1 min
7. Hold at 4°C

We then added 100 µL of binding buffer from the ThermoScientific GeneJET Gel Extraction Kit (#K0692) to each PCR reaction, and performed a standard PCR purification protocol in one column per sample. In the final step, we eluted into 80 µL of elution buffer.

For the second step of PCR, we use standard Nextera XT Index v2 primers (Illumina #FC-131–2004) to further label samples representing different conditions and timepoints with unique identifiers that allow for multiplexing on the same sequencing lane. We uniquely dual-indexed each sample using our nested scheme (see *Mitigating the effects of index hopping* section for details). We performed three reactions of the second step PCR per sample, using the master mix:

- 1.5 µL Q5 Polymerase (NEB #M0491L)
- 30 µL Q5 Buffer (NEB #M0491L)
- 3 µL 10 mM dNTP (Fisher Scientific #PR-U1515)
- 6.25 µL i7 Nextera XT Primer ('N' primer)
- 6.25 µL i5 Nextera XT Primer ('S' primer)
- 78 µL purified step 1 PCR product
- 25 µL Nuclease Free Water (Fisher Scientific #AM9937)

This master mix was then divided into three PCR tubes per reaction, and run with the following protocol on a thermocycler:

1. 98°C for 30 s
2. 98°C for 10 s
3. 62°C for 20 s
4. 72°C for 30 s
5. Repeat steps 2–4 at least 21 times and at most 27 times (for a total of 22 – 28 cycles)
6. 72°C for 3 min
7. Hold at 4°C

We then added 100 µL of binding buffer from the ThermoScientific GeneJET Gel Extraction Kit and purified the PCR product, eluting into 43 µL. We found that increasing the number of cycles in the second step PCR beyond 21 did not seem to improve the amount of DNA recovered after gel extraction. For some samples, we experimented with a touch down procedure for the second step PCR where we started with a hotter annealing temperature and slowly decreased it over the course of 27 cycles. This also did not seem to increase the yield of DNA recovered from the PCR.

## Removal of the reference strain via digestion and gel purification

To avoid the vast majority of our sequencing reads mapping only to the reference strain (and thus not being informative to relative fitness of the mutants), we use restriction digest to cut the ApaLI restriction site in the middle of the reference strain's barcode region. We mixed 43 µL of the second step PCR product with 2 µL of ApaLI (NEB #R0507L) and 5 µL of 10X Cutsmart and incubated at 37°C for at least 2 hr (up to overnight). After digestion, we conducted size selection by running the

digested sample on a gel, removing all product less than 300 bp, and isolating the DNA using a standard ThermoScientific GeneJET Gel Extraction protocol. Our expected product is 350 bp. We did not remove longer sequences via gel extraction because of the possibility that some barcode sequences may selectively form complexes with themselves or other barcodes.

Note that for some samples, we also digested the reference strain before PCR, in addition to after PCR, to decrease the amount of reference strain barcode. For these samples, we mixed 80 µL of genomic DNA (at concentration 50 ng/µL) with 10 µL of 10X Cutsmart and 2 µL of ApaLI and incubated 37°C for at least 2 hr (up to overnight). This product was then used as the template for PCR step 1 (with appropriate water volume adjustments to ensure 50 µL reactions).

## Sample pooling and amplicon sequencing

We used the Qubit High Sensitivity (ThermoFisher #Q32854) method to quantify the concentration of the final product for each sample, then pooled samples with different dual indices in equal frequency for sequencing. Our samples were then sent to either Novogene (https://en.novogene.com/) or Admera Health (https://www.admerahealth.com/) for quality control (qPCR and either Bioanalyzer or TapeStation) and sequencing. We used 2 × 150 paired-end sequencing along with index sequencing reads on Illumina HiSeq machines using patterned flow cells (either HiSeq 4000 or HiSeq X). We also used Illumina Nextseq machines with unpatterned flow cells. We found that the former was more subject to index hopping errors, please see *Mitigating the effects of index hopping* for a discussion of how our dual indexing reduces effects of index hopping. All amplicon samples were sequenced with at least 20% genomic DNA spiked in (either whole genomes from an unrelated project or phi-X) to ensure adequate diversity on the flow cell.

## Mitigating the effects of index hopping

To reduce the effects of index hopping observed on Illumina patterned flow cell technology (including HiSeq 4000, HiSeq X, and Novaseq machines) (*Illumina, 2017*; *Sinha et al., 2017*), we devise a nested unique-dual-indexing approach. This approach uses a combination of inline indices attached during the first step of PCR, as well as Nextera indices attached during the second step of PCR. The latter indices are not part of the sequencing read (they are read in a separate Index Read). This process uniquely labels both ends of all DNA strands such that DNA strands from multiple samples can be multiplexed on the same flow cell. Had we only labeled one end of each DNA strand, index hopping could have caused us to incorrectly identify some reads as coming from the wrong sample.

One approach to label samples with unique-dual-indices is to use 96 forward primers, each of which is paired to one of 96 reverse primers, instead our nested approach allows us to uniquely dual-index samples with only 40 total primers (12 forward inline, eight reverse inline, 12 Nextera i7, 8 Nextera i5). Specifically, we can use combinations of the Nextera and inline primers. One way to think of this is that there are 96 possible ways to combine the forward inline and Nextera i5 primers that are on the same side of the read, effectively creating 96 unique labels for that end of the read.

To reduce the effect of index hopping contamination on our results, we included only samples that were sequenced on non-patterned flow cell technology (HiSeq 2000 and 2500 for samples in batches 1–6, 10, NextSeq for samples in batch 9) or were sequenced on patterned flow cell technology (patterned flow cell HiSeq) with nested unique-dual indexing.

## Processing of amplicon sequencing data

We processed the amplicon sequencing data by first using the index tags to de-multiplex reads representing different conditions and timepoints. Then, using Bowtie2 (*Langmead and Salzberg, 2012*), we mapped reads to a known list of barcodes generated by *Venkataram et al., 2016a*, removed PCR duplicates using the UMIs from the first-step primers, and counted the number of reads for each barcode in each sample. The source code for this step can be found at *Venkataram, 2020*. We processed all raw data for this study using this pipeline, including re-processing the raw sequencing files for data from previous studies (*Li et al., 2018*; *Venkataram et al., 2016a*) so that all data was processed together using the most recent version of the code.

Several samples included technical replicates where the sample was split at various times in the process, including before DNA extraction, before PCR, and prior to sequencing. Read counts across these technical replicates were merged in order to calculate the best estimate of barcode

frequencies. Counts were merged after appropriately accounting for PCR duplicates as identified from Unique Molecular Identifiers.

## Quantification and statistical analysis

### Fitness estimate inference

The amplicon sequencing data shows the relative frequency of each barcode in each time-point of every one of our 109 fitness measurement experiments. To estimate the fitness of each barcoded mutant in each experiment, we calculate how barcode frequencies change over time. We do this using previously described methods (*Venkataram et al., 2016a*).

Briefly, we first calculated the log-frequency change of each barcoded adaptive mutant for each subsequent pair of time-points. This log-frequency change must be corrected by the mean fitness of the population, such that it represents the relative fitness of each mutant relative to the reference strain, which makes up the bulk of the population. Since we destroyed barcodes pertaining to the reference strain by digesting them, we infer how the mean fitness of the population changes at each time-point using barcoded lineages that are known to be neutral (see *Identification of neutral lineages*). Once we calculated the change in the relative fitness of each barcoded mutant across each pair of consecutive time-points, we took a weighted average across all pairs as our final estimate of each adaptive mutant's relative fitness for a given experiment. We weighted each pairwise fitness estimate using an uncertainty measure generated from a noise model (see *Noise model* section below).

This results in 109 fitness measurements per each barcoded mutant, with some of the 45 conditions having more representation than others due to having more replicates. In cases where we have replicates, we averaged the fitness values across the replicates, weighted by the measurement uncertainty, resulting in our final 45 fitness estimates per each adaptive mutant lineage.

We aimed to sequence each timepoint at a depth of at least 100X coverage per barcode (totaling ~50,000 reads that map to barcodes that are not the ancestral barcode). Over 95% of all timepoints have coverage in excess of this target, with ~70% of the timepoints exceeding 500,000 mapped reads (~1000X coverage per barcode). In order to include as many conditions and timepoints as possible, we included lower coverage timepoints that had at least 2500 mapped reads as long as at least 400 barcoded mutants were represented. Because our noise model accounts for uncertainty due to read depth, these timepoints are under-weighted when calculating the overall fitness across all four or five timepoints. There were two conditions we included whose average coverage across included timepoints was below the target of 100X coverage per barcode. The Baffle, 2.5% Glucose condition had average mapped read counts per timepoint of 33918 and 17189 for replicates 1 and 2, respectively. The Baffle, 0.4 µg/ml Benomyl condition had average mapped reads counts per timepoint of 15,381 and 15,077 for replicates 1 and 2, respectively. However, despite having coverages lower than the target, there is reasonable agreement between the replicates (correlation coefficient $r = 0.57$ and $r = 0.77$, respectively). Additionally, both these environments are classified as strong perturbations from the evolution condition, so their inclusion does not affect the inference of the phenotype model. Finally, including these conditions is conservative in that any noise in their fitness estimate would make it more difficult to predict fitness in these environments.

### Identification of neutral lineages

Previous work using this fitness measurement method focused on a larger collection of 4800 barcoded yeast lineages, where the vast majority of these lineages were neutral (*Li et al., 2018*; *Venkataram et al., 2016a*). In order to increase the number of reads per adaptive lineage, we used a smaller pool of 500 lineages for most experiments. However, this prevents us from identifying neutral lineages as was done in previous studies, by rejecting outlier lineages with higher than typical fitness values. Instead, we used a set of 35 high-confidence neutral lineages to infer mean fitness (see *Experimental model and subject details*). These lineages showed no fitness differences from the neutral expectation in previous studies and were shown to possess no mutations detectable via whole genome sequencing. These high-confidence neutral lineages were present in all experiments, and were spiked into experiments from batch nine to increase their frequency. We used these neutrals to perform the fitness inference in two steps. First, we inferred fitness using this collection of high-

confidence neutrals to make a first pass at inferring the fitness values. Next, we included lineages with similar behavior to the high-confidence neutrals to improve our estimate of mean fitness.

## Noise model

To quantify the uncertainty for each fitness measurement, we used the noise model as outlined in *Venkataram et al., 2016a*.

Briefly, this noise model accounts for the uncertainty coming from several sources of noise. The first type of noise scales with the number of reads for a given lineage. This noise stems from stochasticity in population dynamics (coming from the inherent stochasticity in growth and noise associated with dilution), from counting noise associated with a finite coverage, and technical noise from DNA extraction and PCR. We fit this noise by quantifying the variation in the frequency of neutral lineages (see *Identification of neutral lineages*). There is additional variation in fitness observed for high-frequency lineages between replicate experiments (here we refer to variation across replicates that were performed simultaneously, not variation across batches). We also accounted for this uncertainty following previous studies. Specifically, we fit an additional frequency-independent source of noise using between-replicate variation.

## Checks on noise model

Because our ability to count the phenotypes that matter to fitness hinges upon measurement error, we further assessed the accuracy of our noise model. We did so by using barcoded lineages that should have the same fitness because they are genetically identical. Since our fitness estimates are imperfect (i.e. they contain some noise), we estimated each of these lineages as having slightly different fitness. We then asked if the variation in fitness across these lineages is explained by our noise model, or if there is more variation than our noise model can account for. We did this explicitly by calculating, for each lineage, how far our fitness estimate is from the best guess for the true underlying fitness value (the group's mean) in units of the estimate's measurement precision. We then calculated the percent of lineages that are a given distance from the group's average to understand the accuracy of the model. For instance, if the noise model perfectly captures the uncertainty of each measurement, then 10% of the diploid lineages should have a difference from the weighted diploid mean in the 10th percentile, 20% in the 20th percentile, etc. Because 188 of our 292 barcode mutants are diploids without additional mutations, diploids are an ideal group to use to assess the accuracy of the noise model. This procedure shows that, for the vast majority of replicates, the noise model is conservative. That is, diploid lineages tend to have less variation in fitness than expected by the noise model (*Figure 2—figure supplement 1*).

## Classifying mutants by mutation type

Some types of mutants are present more than others. For example, 188 of our 292 mutants are diploids and 30 mutants are in the IRA1 gene. If not properly accounted for, this imbalance can lead to some unfairness in predictions for our model. For example, if we use mostly diploid lineages to train our model, we will be very good at predicting the fitness of diploids but poor at predicting other types of mutants. This means that we must classify our mutants by mutation type in order to properly balance them. We classified mutants following previous work (*Venkataram et al., 2016a*) that classified mutants as either diploids, or if haploid, by the gene possessing the putative causal mutation. Because previous work finds differences in fitness between missense and nonsense/frameshift/indel mutations in IRA1, here we classified these mutants into 'missense' and 'nonsense' classes, where mutants with frameshift and indel mutations were classified as 'nonsense'. We also classified diploid mutants with additional mutations in nutrient-response genes or chromosomal amplifications as separate groups. Additionally, we created a separate class for 'high-fitness diploid' mutants that possess no additional detected mutations (other than being diploid) but have very high fitness in the EC. To be classified as a high-fitness diploid, a diploid mutant must have an average fitness across all nine EC batches that is greater than two standard deviations above the average of all diploids. In the main text, we label these mutants as 'diploid with additional mutation' since they are likely to harbor additional mutation(s) due to their increased fitness.

## Calculation of weighted average Z score

To partition environments into subtle and strong perturbations of the EC, we relied on the nine experiments performed in the EC. As each of these experiments was performed at a different time, variation in fitness across these experiments represents batch effects, and we therefore refer to these nine experiments as 'EC batches'. Environmental differences between batches are very subtle, as they represent the limit of our ability to minimize environmental variation. Thus, variation in fitness across the EC batches serves as a natural benchmark for the strength of environmental perturbations. If the deviations in fitness caused by an environmental perturbation are substantially stronger than those observed across the EC batches, we call that perturbation 'strong'.

More explicitly, to determine whether a given environmental perturbation is subtle or strong, we first quantified the typical variation in fitness for each mutant, across the EC batches:

$$\sigma_i = \frac{1}{n_{batches}} \sum_j^{batches} |f_{ij} - \bar{f}_i|$$

where $\sigma_i^2$ represents the variance in fitness across the EC batches for mutant $i$, and $\bar{f}_i$ represents the average fitness of mutant $i$ across the EC batches.

To ensure that each mutation type contributes equally to our classification of how different each environment is from the evolution condition, we weighed each mutant's contribution to this difference. We did so based on the number of mutants with the same mutation type, such that the mutation-type-weighted average Z-score for a given environment $j$ is given by:

$$z_j = \sum_i^{mutants} \frac{|f_{ij} - \bar{f}_i|}{n_{type(i)}\sigma_i}$$

where $n_{type(i)}$ represents the number of mutants that are the same mutation type as mutant $i$.

We then classified the environmental perturbations based on this Z-score. Sixteen environments provoked fitness differences resulting in a Z-score of less than two, and we classified these environmental perturbations as 'subtle'. The remaining 20 environments had Z-scores greater than 2, which we classified as 'strong' environmental perturbations.

## Model of phenotypes that contribute to fitness

In order to count the phenotypes that affect fitness in our collection of mutants, we explored a low-dimensional phenotypic model. We explicitly used a model of fitness-relevant phenotypes such that each mutant is represented as having a fixed effect on each phenotype, represented by a vector of $k$ phenotypes, for example mutant $i$ is represented by the vector $(p_{i1}, p_{i2}, p_{i3}, ..., p_{ik})$. In addition, each environment is represented by a vector of phenotypic weights, representing the importance of each of the $k$ phenotypes to fitness in that environment, for example environment $j$ represented by the column vector $(e_{1j}, e_{2j}, e_{3j}, ..., e_{kj})$. The fitness effect of mutant $i$ in a given environment $j$ is the linear combination of that mutant's phenotypes, each weighted by its importance in environment $j$:

$$f_{ij} = p_{i1}e_{1j} + p_{i2}e_{2j} + p_{i3}e_{3j} + ... + p_{ik}e_{kj}$$

Our fitness measurements reflect mutant fitness relative to a reference strain, therefore, our model places the reference strain (which has fitness 0 by definition) at the origin of this multi-dimensional space. Our model only includes phenotypes that differ between the reference strain and least one mutant. This is sensible given that our reference strain is a modified version of the ancestor of all these mutant lineages. Thus, if there exists a phenotype that contributes to fitness, but none of the adaptive mutants altered that phenotype, our model will not detect it. More explicitly, a phenotype that contributes to fitness would have a non-zero value of $e$, but if no mutant alters that phenotype from the reference, all mutants would have a zero value of $p$ for that phenotype. Thus, the non-zero value of $e$ would always be multiplied by a zero value for $p$ and this phenotypic dimension would not be represented in our model. This is not to say that if only a single mutant of the 292 alters a particular phenotype we would include it as a phenotypic dimension. Our power to add dimensions to our model is limited by measurement noise. We only include dimensions that capture

more variation in fitness than do dimensions that capture measurement noise (see *Estimating the detection threshold using measurement error*).

Similarly, because we measure fitness, and not phenotype, our model is blind to any phenotypic effect that does not contribute to fitness in at least one of the 45 environments we studied. If a mutant has large phenotypic effects, but they do not cause that mutant's fitness to differ from the reference strain in any of these 45 environments, this phenotypic effect will not be represented in our low-dimensional phenotypic model. More explicitly, mutants may have non-zero phenotypic effects *p*, but if these do not influence their fitness in any environment we study, *e* will be zero for all 45 environments. Thus, *p* times *e* will also be zero and we will not include this phenotypic dimension in our model.

Importantly, the phenotypic dimensions that we infer from our fitness measurements are abstract entities. They represent causal effects on fitness, rather than measurable features of cells. For this reason, they might be called 'fitnotypes' (a mash of the terms 'fitness' and 'phenotype'). Even though the fitnotypes are independent with respect to their contribution of fitness, and contribute to fitness linearly, the mapping of commonly measured features of cells (e.g. growth rate, the expression levels of growth supporting proteins like ribosomes) onto fitnotypes may be more complicated. For instance, a commonly measured cellular feature that has a complicated nonlinear mapping to fitness could be detected as many, linearly contributing fitnotypes. This is another reason that our phenotypic dimensions are not necessarily comparable to what people traditionally think of as a 'phenotype'.

## Using SVD to decompose the fitness matrix

Our goal is to use fitness measurements to learn about the phenotypic effects of mutations as well as the contribution of these phenotypes to fitness in different environments. We conducted fitness measurements for 292 mutants in each of 45 environments and organized these data into a fitness matrix, *F*, where every row corresponds to a mutant, every column corresponds to an environment, and every entry is a fitness measurement. Because our model (see *Model of phenotypes that contribute to fitness*) represents fitness in a given environment as the sum of multiple phenotypes, each scaled by their contribution to fitness in that environment, we can use SVD to decompose the fitness matrix *F* as:

$$P\Sigma E^T = F$$

The left hand side of this equation consists of three matrices: *P*, which represents the positions of the mutants in our low-dimensional model of phenotypic space, $E^T$, which represents the contribution of a phenotype to fitness in a given environment, and Σ, a diagonal matrix representing the singular values of the fitness matrix *F*. Though the singular values are informative in this separation of three matrices, particularly for the amount of variation captured by each of the inferred components, we can also think of this as a decomposition into two matrices, where we fold the singular values into either the mutant phenotypes or the environment weights, as described in the main text. Either way, this decomposition captures the data represented in the fitness matrix *F*, including measurement error as well as the underlying biological signals.

Importantly, the dimensions in the phenotypic model we built using SVD are detected in the order of their explanatory power. Moreover, the first dimension is the best, linear one-component model that explains the data (if evaluated by mean squared error). This is true for any set of the first *k* components. This means, for example, that the model with the first eight components is the best possible eight-component linear model for the observed data (*Eckart and Young, 1936*).

One issue in this type of analysis is that adding more components always improves the explanatory power of the model, even when those components capture variation that is primarily due to measurement noise. This type of overfitting problem is common in statistics, and several methods have been devised to select the appropriate number of components to include. We use two such methods here.

## Estimating the detection threshold using measurement error

One method to select the appropriate number of components to include in the model and prevent overfitting (i.e. prevent fitting a component that primarily represents noise) is to use measurement

error as a type of control. This is only possible if the amount of measurement error is known. We estimated the amount of noise in our fitness measurements using a previously described noise model (see *Noise Model*) (*Venkataram et al., 2016a*). Since this noise model includes counting noise, every fitness measurement may have a different amount of noise. For example, mutants present at low frequency will be subject to more stochasticity resulting from counting noise. We used this noise model to simulate fitness tables (*F*) where mutant fitnesses vary exclusively due to measurement noise. We simulated 1000 noise-only matrices, where each entry is pulled from a normal distribution centered at zero and with variance equal to the estimated measurement noise of the corresponding entry in the true fitness matrix *F*. We then applied SVD to each noise-only matrix, which gave us a set of singular values generated only by noise. From many such simulations, we took the average size of the largest component, which reveals how much variation can be explained by a component that captures only noise. We found that the largest noise-components are of the size that they would capture 0.07% of variation in our true fitness matrix. Thus, we set this as our limit of detection. In other words, in order for us to include eight components in our low-dimensional model, all of them must explain more than 0.07% of the variation in fitness. This approach is analogous to identifying a threshold when measurement noise is known but not identical for all entries in the matrix (*Josse and Sardy, 2014*).

## Estimating detection threshold using bi-cross-validation

Another method for identifying the appropriate number of components is to use their predictive power. This method relies on the intuition that measurement error is uncorrelated across different mutants and different environments. Therefore, a component that represents measurement error should not contain information that can help predict the fitnesses of these mutants in new environments. It should also not contain information that can help predict the fitness of unstudied mutants. We used a bi-cross-validation scheme of the SVD devised by *Owen and Perry, 2009* which divides the mutants and environments into distinct groups of training and testing sets. This subsequently divided our matrix of fitness measurements into four submatrices: the fitness of the training mutants in the training environments (*D*), the fitness of the training mutants in the testing environments (*C*), the fitness of the testing mutants in the training environments (*B*), and the fitness of the testing mutants in the testing environments (*A*).

$$F = \begin{pmatrix} A \begin{smallmatrix} \textit{Test Mutants} \\ \textit{Test Environments} \end{smallmatrix} & B \begin{smallmatrix} \textit{Test Mutants} \\ \textit{Train Environments} \end{smallmatrix} \\ C \begin{smallmatrix} \textit{Train Mutants} \\ \textit{Test Environments} \end{smallmatrix} & D \begin{smallmatrix} \textit{Train Mutants} \\ \textit{Train Environments} \end{smallmatrix} \end{pmatrix}$$

We carried out SVD on the training data (submatrix *D*), which returned a set of singular values and corresponding components that captured the fitness data in *D*. We then used these components to predict the fitness of the testing mutants in the testing environments (submatrix *A*). First, we tried to predict these fitness values by only using the first component. That is, we fixed this first component and the first singular value for the training mutants. We then found the best first component for the testing environments based on the fitness values of the training mutants in these environments (i.e. using the information in submatrix *C*), given the constraint that the training mutants can only be represented by the one component. We then conducted an analogous procedure to find the first component of the testing mutants by fixing the first component of the training environments by using the information in submatrix *B*. Then, we tried to predict the fitness of the testing mutants in the testing environments using the first component independently fit for each. We subsequently repeated this procedure, giving the testing mutants access to more of the training components each time. If the components detected by the training components represent biological signal, then this should improve the ability to predict the fitness of the testing mutants in the testing environments. However, once the components primarily represent measurement error, their inclusion should harm predictive power. Therefore, we use the number of components with the best ability to predict the held-out data (submatrix *A)* as the number of components that represent biological signal in our data.

For computational efficiency, we explicitly used the formulation proposed by *Owen and Perry, 2009* for the prediction of the held-out submatrix *A*:

$$\hat{A} = B \left( \hat{D}^{(k)} \right)^{+} C$$

where $(\hat{D}^{(k)})^+$ denotes the Moore-Penrose inverse of the rank $k$ approximation of sub-matrix $D$. This prediction is equivalent to the procedure outlined above, provided that least-squares regression is used to identify the components of the testing mutants and testing conditions, conditional upon the training components (*Owen and Perry, 2009*).

We divided our mutants into fixed training and testing sets (see *Division of Mutants into Training and Testing Sets*) and used these sets throughout our study. As for training versus testing environments, these changed depending on our goal. For validating the number of components to include in our phenotypic model, we held out each of the 25 subtle environmental perturbations, using it as the testing environment and the other 24 for training. For making predictions of the fitness of the testing mutants in the strong environmental perturbations, we used all 25 subtle environmental perturbations as the training set, though we also show how these predictions vary when each of the 25 subtle environmental perturbations is held out from the training set.

## Division of mutants into training and testing sets

In order to perform bi-cross-validation on our data, we need to divide our data into training and testing sets. Because some mutation types, in particular diploids and Ras/PKA mutants, are present more than others in our collection of mutants, we sampled the training set such that each mutation type is represented roughly equally (see *Classifying mutants by mutation type*). Specifically, we designated half of each mutation type, with a maximum of 20 representatives of each type, as belonging to the training set. The remaining mutants comprise the test set. For example, there are 188 diploids included in the 292 adaptive mutants. We included 20 in the training set and 168 in the test set. There are 20 *IRA1 nonsense* mutants included in the 292, and we included 10 in the training and 10 in the test set. Additionally, genes that are represented only once in the set of mutations are placed in the test set. This results in a training set of 60 mutants and a testing set of 232 mutants (see *Supplementary file 1*).

## Using simulated data to validate detection threshold estimation

To further validate our approach for identifying the number of detectable phenotypic components from our data, we simulate data that consists of a known number of phenotypic components $k$ and use our methods to estimate the number of phenotypic components detectable in the data. To simulate the phenotype space, we place 100 mutants at random in the $k$-dimensional phenotype space $P$. The coordinates of these mutants are pulled from a uniform distribution in the n-ball (e.g. the n-ball is a sphere if there are three dimensions) centered at coordinates (1, 0, . . ., 0) with radius 1. We center the mutants at one in the first dimension and 0 in all other dimensions in order to create data similar to our empirical data where the first component captures much of the variation in fitness. We then similarly place 50 environments at random in the $k$-dimensional environmental space $E$. Recall that this space represents the importance of each phenotype in each environment (see *Figure 3* and see *Model of phenotypes that contribute to fitness*). The environments are pulled from a uniform distribution in the n-ball centered at (1, 1,..., 1) with a small radius of 0.1 chosen such that the environmental perturbations are subtle. Note, for computational efficiency, we use the algorithm from *Marsaglia, 1972* to pull points uniformly distributed in the n-ball. Next, we calculate the fitness of each of these mutants in each environment as a linear combination of the mutant's phenotypes weighted by the contribution of each of these phenotypes to fitness in the relevant environment (see *Model of phenotypes that contribute to fitness*). We then add measurement error to these fitness values to simulate the effect that measurement uncertainty has on our ability to detect phenotypic components. We simulate the data with various numbers of phenotypic components (2, 3, 4, 5, 10, 20, 30, 40, and 49) and use our methods to try to estimate the number in each set.

We find that our method for identifying the number of detectable phenotypic components from the measurement error (see *Estimating the detection threshold using measurement error*) accurately identifies the simulated number of phenotypic components when measurement noise is very low (*Figure 3—figure supplement 1A*). As measurement noise increases, our approach detects fewer components, as expected due to measurement noise swamping the smallest components of signal (*Figure 3—figure supplement 1A*). Bi-cross-validation, which holds out each environment and half of the mutants (see *Estimating the detection threshold using bi-cross-validation*), performs similarly,

detecting the appropriate number of phenotypic components when measurements are sufficiently precise (*Figure 3—figure supplement 1B*).

## Clustering mutants in phenotype space

After inferring the low-dimensional model of phenotype space using SVD, we used Uniform Manifold Approximation and Projection (UMAP) to visualize how the mutants cluster in that space. For this analysis, we used the eight-component phenotypic model that we built from the 60 training mutants and the 25 subtle perturbations. We did this to avoid the model being dominated by variation in very common mutations, specifically the diploids, which make up 188/292 of our adaptive mutants. We added more mutants in the visualization by finding the location of each of the testing mutants (except diploids) by least sum of squares optimization. To do so, we fixed the coordinates for the 25 environments and found the coordinates for each mutant that best estimated its fitness in all environments. To further avoid our visualization being dominated by the diploids, we included only the diploids present in the training set in our visualization. For UMAP, we specified that 20 neighbors are used.

Although UMAP tends to preserve both local and global structure (*McInnes et al., 2018*) it is not necessarily representative of the distance between objects in high-dimensional space. Thus, to quantify more precisely the clustering by gene observed, we explicitly compared the median pairwise distance between these apparent clusters to 10,000 randomly chosen sets of the same size and calculated empirical p-values. Because there are many diploids such that they will be the most prevalent type of mutant drawn in these randomly chosen sets, we only drew from strains that have other mutations besides or in addition to diploidy. We use the median pairwise distance, rather than the mean, to identify the typical distance between mutants in a given cluster to reduce the influence outlier mutations that might bias the mean pairwise distance.

## Calculation of weighted coefficient of determination

Because mutants are present in unequal numbers in the test set, standard measures of variance explained are likely to be representative of our ability to predict mutants that have many barcoded lineages present in the data, for instance diploid and *IRA1 nonsense* mutations. These measures would be less representative of mutants with few lineages present, that is TOR/Sch9 pathway mutants. Thus, we use a measure of predictability ($\tilde{R}^2$) that weights the contribution of each mutant to overall variance explained based on the number of lineages that share its mutation type (diploids, *IRA1 nonsense*, *IRA1 missense*, *GPB2*, etc.). This effectively measures our ability to predict the fitness of each mutation type, rather than each mutant. For overall predictive power across all mutants and conditions, we used the measure:

$$\tilde{R}^2 = 1 - \frac{\sum_i^{mutants} \sum_j^{conditions} \frac{1}{n_{type(i)}} \left(f_{ij} - \hat{f}_{ij}\right)^2}{\sum_i^{mutants} \sum_j^{conditions} \frac{1}{n_{type(i)}} \left(f_{ij} - \bar{f}\right)^2}$$

where $\bar{f}$ denotes the average fitness for all evaluated mutants and evaluated conditions.

We used a similar measure to quantify the ability to predict fitness for each environment *j*. This is given by:

$$\tilde{R}_j^2 = 1 - \frac{\sum_i^{mutants} \frac{1}{n_{type(i)}} \left(f_{ij} - \hat{f}_{ij}\right)^2}{\sum_i^{mutants} \frac{1}{n_{type(i)}} \left(f_{ij} - \bar{f}_j\right)^2}$$

where $\bar{f}_j$ denotes the average fitness across all evaluated mutants in condition *j*.

Note that this measure explicitly compares a model's fitness prediction in each environment to predictions made using the average fitness in that environment, such that if the model's fitness prediction is the same as the average fitness, $\tilde{R}^2$ is zero. It is possible that a given model's fitness prediction is worse than that of the average fitness in that environment, resulting in negative values of $\tilde{R}^2$. In our work, negative values occur for the one-component model when predicting the fitness of mutants in some of the strong environmental perturbations. In particular, this occurs when fitness in

that environment is uncorrelated with EC fitness, which is captured by the first component, such that the EC fitness is unable to make reasonable predictions of fitness in this environment.

Note that we observe qualitatively similar results to this measure when we use a standard variance explained measure and exclude diploids, which dominate the test set (see *Figure 4—figure supplement 4*).

## Evaluating the effect of the number of environments and mutations used

Because the detection of phenotypic components is dependent on the choice of environments and mutations used, it is important to understand how these influence the number of components we detect and our ability to predict fitness in held-out data.

To evaluate how the number of subtle environments included in the training of the phenotype model affects the number of detected components and predictive power, we performed the following analyses. We used SVD to infer phenotypic models using randomly-selected subsets of 25 subtle environmental perturbations, which ranged in size from 10 to the full number of subtle environmental perturbations, 25. We repeated this 25 times for each subset size, ensuring that the subsets were not identical. We found that the median number of detected phenotypic components increases with the number of subtle environments included in the subset. Specifically, we detected only five components when only 10 subtle environmental perturbations are used (down from eight components when the full set of 25 subtle environmental perturbations are used). In addition, we find that the model's ability to predict the fitness of the test mutants in the strong environmental perturbations decreases with subset size. In other words, when we use fewer subtle environments to build the model, the model has less predictive power. This prediction power ranges from explaining on average 65% of weighted variation when we use only 10 subtle environments to build the model, to 74% of weighted variation explained with the full 25 subtle environments (*Figure 4—figure supplement 1*).

To understand how the number of mutations included in the training set affects the number of detected components, we used SVD to infer phenotypic models using the full set of subtle environmental perturbations and random subsets of the 60 training mutants. We found that as we increased the number of mutants in the training set, the number of components and the ability of the model to predict the fitness of test mutants in the strong environmental perturbations increased, ranging from explaining on average 65% of weighted variance with only 10 mutants in the training set to explaining 74% of weighted variance with the full training set. To test if this pattern was primarily driven by the number of mutants or instead the number of distinct mutation types, we repeated this process, instead subsampling random sets of mutation types (see *Classifying mutations by mutation type*). Here, we find a strong relationship between the number of mutation types included in the training set and the amount of weighted variance predicted for the test mutants in the strong environmental perturbations. We explain on average 50% of weighted variance with only two mutation types, compared to the full 74% of weighted variance with the full training set (*Figure 4—figure supplement 2*). In addition, despite this general trend that the inclusion of more mutation types increases the ability to predict fitness of held-out data, there is relatively little variation in overall predictive accuracy when any single mutation type is excluded, suggesting the model's predictive accuracy is robust to the inclusion of any particular mutation type in the training set.

## Calculating mutant-specific improvement

It is possible that all 292 of our adaptive mutants each affect all eight of the phenotypic components in our low-dimensional model; however, it is also possible that some mutants influence some phenotypes more strongly than others. In order to quantify how much a specific component lends to the ability to predict the fitness of each mutant in each environment, we need a metric to calculate the difference in predictive accuracy for the model with and without this component. Specifically, to assess the impact of the inclusion of the $k$th component, we compared the prediction accuracy of the $k$-component model to the model that includes the first $k-1$ components.

Because fitness estimates vary in their reliability due to finite coverage and other sources (see *Noise model* section), we should factor this uncertainty in our measure of prediction improvement. For example, a small improvement in prediction accuracy for a very uncertain fitness estimate is less meaningful than the same improvement in prediction accuracy for a fitness estimate that we are

quite confident in. Thus, we scale the difference in prediction accuracy by the amount of uncertainty in the underlying fitness estimate.

This gives us the measure of improvement in the estimate of the fitness of mutant $i$ in condition $j$ due to the inclusion of the $n$th component as:

$$I_{ij}^k = \frac{\left(\hat{f}_{ij}^{\,k-1} - f_{ij}\right) - \left(\hat{f}_{ij}^{\,k} - f_{ij}\right)}{\epsilon_{ij}}$$

where $\hat{f}_{ij}^{\,k}$ and $\hat{f}_{ij}^{\,k-1}$ represent the estimate of the fitness of mutant $i$ in condition $j$ for the model with $k$ and $k$-1 components, respectively. $f_{ij}$ and $\epsilon_{ij}$ represent the measured fitness value and measurement uncertainty for the fitness of mutant $i$ in condition $j$, respectively.

## Data and code availability

### Data resource
The raw Illumina sequencing data for the fitness measurement assays conducted in this study can be found under NIH BioProject: PRJNA641718. Sequencing data previously published in *Venkataram et al., 2016a* can be found under NIH BioProject: PRJNA310010. Sequencing data previously published in *Li et al., 2018* can be found under NIH BioProject: PRJNA388215.

### Code
The software repository for the barcode counting code can be found at *Venkataram, 2020*.

The software repository for the fitness estimate inference can be found at *Venkataram et al., 2016b*.

The code for all downstream analysis, including figure generation can be found at *Kinsler et al., 2020*.

## Acknowledgements
The authors thank Sandeep Venkataram for the BarcodeCounter2 script; Yuping Li, Monica Sanchez, Tuya Yokoyama, Chris McFarland, Grace Lam, Ellie Armstrong, and Dimitra Aggeli for technical assistance; Atish Agarwala, Joy Bergelson, Marc Salit, Sasha Levy, Gavin Sherlock, Ben Good, Ivana Cvijovic, David Gokhman, Emily Ebel, Simon Levin, Molly Schumer, Jan Skotheim, Moises Exposito-Alonso, Mikhail Tikhonov, Hunter Fraser, Michael Desai and all members of the Petrov and Geiler-Samerotte Labs for helpful comments and discussions. We are grateful to the twitter community that followed #1BigBatch and provided us with very helpful feedback. We are grateful to Enrico Coen for very helpful discussions and specifically for the suggestion of the term 'fitnotype'. Some of the computing for this project was performed on the Sherlock cluster. We would like to thank Stanford University and the Stanford Research Computing Center for providing computational resources and support that contributed to these research results. This work was supported by National Institutes of Health grant R35GM118165 (to DAP) and National Institutes of Health grant R35GM133674 (to KGS).

## Additional information

### Funding

| Funder | Grant reference number | Author |
| --- | --- | --- |
| National Institutes of Health | R35GM118165 | Dmitri A Petrov |
| National Institutes of Health | R35GM133674 | Kerry Geiler-Samerotte |

The funders had no role in study design, data collection and interpretation, or the decision to submit the work for publication.

## Author contributions
Grant Kinsler, Kerry Geiler-Samerotte, Conceptualization, Resources, Data curation, Software, Formal analysis, Validation, Investigation, Visualization, Methodology, Writing - original draft, Project administration, Writing - review and editing; Dmitri A Petrov, Conceptualization, Resources, Supervision, Funding acquisition, Writing - original draft, Project administration, Writing - review and editing

## Author ORCIDs
Grant Kinsler (iD) https://orcid.org/0000-0001-8308-4665
Kerry Geiler-Samerotte (iD) https://orcid.org/0000-0003-4666-2192
Dmitri A Petrov (iD) https://orcid.org/0000-0002-3664-9130

## Decision letter and Author response
Decision letter https://doi.org/10.7554/eLife.61271.sa1
Author response https://doi.org/10.7554/eLife.61271.sa2

# Additional files

## Supplementary files
• Supplementary file 1. List of all mutants included in this study.

• Supplementary file 2. List of all conditions used in this study, ordered by deviation from the EC batch as in the main text figures.

• Transparent reporting form

## Data availability
All sequencing data has been deposited to SRA under NIH BioProject number PRJNA641718.

The following dataset was generated:

| Author(s) | Year | Dataset title | Dataset URL | Database and Identifier |
|---|---|---|---|---|
| Kinsler G, Geiler-Samerotte K, Petrov D | 2020 | A genotype-phenotype-fitness map reveals local modularity and global pleiotropy of adaptation | https://www.ncbi.nlm.nih.gov/bioproject/PRJNA641718/ | NCBI BioProject, PRJNA641718 |

The following previously published datasets were used:

| Author(s) | Year | Dataset title | Dataset URL | Database and Identifier |
|---|---|---|---|---|
| Venkataram S, Dunn B, Li Y, Agarwala A, Chang J, Ebel E, Geiler-Samerotte K, Hérissant L, Blundell JR, Levy SF, Fisher DS, Sherlock G, Petrov DA | 2016 | A comprehensive genotype-fitness map of adaptation-driving mutations in yeast | https://www.ncbi.nlm.nih.gov/bioproject/?term=PRJNA310010 | NCBI BioProject, PRJNA310010 |
| Yuping Li, Venkataram S, Agarwala A, Dunn B, Petrov DA, Sherlock G, Fisher DS | 2018 | Understanding *S. cerevisiae* adaptation and trade-offs under batch culture condition | https://www.ncbi.nlm.nih.gov/bioproject/PRJNA388215/ | NCBI BioProject, PRJNA388215 |

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
