## [Decision Letter]

**Acceptance summary:**

The distribution of pleiotropic effects of mutations selected in a particular environment is of broad and fundamental significance. Screens of beneficial genetic variation have taught us that the rising tide of these mutants in the focal environment often lifts other boats in neighboring conditions, but not in orthogonal conditions, where outcomes are unpredictable. This well written, executed, and analyzed study shows that we actually can gain predictability if the number of environments scales to dozens, mutants scale to hundreds, and most importantly, multidimensional analyses are taken seriously enough to derive the most salient predictor variables. The authors find that a low-dimensional phenotypic model is sufficient for capturing fitness of the panel of mutants across subtle environmental perturbations – which suggests that only a few phenotypes contribute to fitness near the evolution conditions. Further, the model accurately predicts fitness in environments that deviate from the evolution condition, often through components that contribute little to fitness near the evolution condition – which suggests that adaptive mutants have latent phenotypic effects that only impact fitness in distant environments.

**Decision letter after peer review:**

Thank you for submitting your article "A genotype-phenotype-fitness map reveals local modularity and global pleiotropy of adaptation" for consideration by *eLife*. Your article has been reviewed by three peer reviewers, including Vaughn S Cooper as the Reviewing Editor and Reviewer #1, and the evaluation has been overseen by Naama Barkai as the Senior Editor. The following individual involved in review of your submission has agreed to reveal their identity: David Gresham (Reviewer #3).

The reviewers have discussed the reviews with one another and the Reviewing Editor has drafted this decision to help you prepare a revised submission.

Summary:

The distribution of pleiotropic effects of mutations selected in a particular environment is of broad and fundamental significance. We've known for a while from large and even larger-scale screens of beneficial genetic variation that the rising tide of these mutants in the focal environment often lifts other boats in neighboring conditions, but not in orthogonal conditions, where outcomes are unpredictable. This well written, executed, and analyzed study shows that we actually can gain predictability if the number of environments scales to dozens, mutants scale to hundreds, and most importantly, multidimensional analyses are taken seriously enough to derive the most salient predictor variables. The authors find that a low-dimensional phenotypic model is sufficient for capturing fitness of the panel of mutants across subtle environmental perturbations – which suggests that only a few phenotypes contribute to fitness near the evolution conditions. Further, the model accurately predicts fitness in environments that deviate from the evolution condition, often through components that contribute little to fitness near the evolution condition – which suggests that adaptive mutants have latent phenotypic effects that only impact fitness in distant environments.

The reviewers took issue with a central claim of the paper, however: a "genotype-phenotype-fitness map" was not constructed as we might normally think, and thus the title is misleading and the goal of such a map was not met. The study does not name and measure the phenotypes themselves; rather it perturbs environmental conditions and measures mutant fitness across environments, which are fitness components that form a collection of abstract phenotypes (as the authors acknowledge) that contribute significantly to fitness.

1) The actual phenotypes need to be defined better if we are to accept the title.

2) The lack of explicit connection of these fitness components (phenotypes) to known genotypes is also a missed opportunity. Please provide examples of genotype-phenotype relationships in a traditional sense, it will broaden understanding.

3) One key analysis missing from the paper involves the degree to which the number of phenotypic dimensions is influenced by the number of environments (and mutants). This could be investigated through construction of new models by iteratively removing environments (or mutants) from the existing dataset to see how the number of dimensions (and their predictive power) changes. Inclusion of this analysis is necessary to see how the phenotypic dimensionality is a function of environments (or mutants) tested.

4) In the manuscript, the authors use the term phenotype to represent an abstract entity that contributes to fitness rather than "an observable trait" as it has been classically defined. This distinction should be apparent in the Abstract or Introduction to allow readers to fully grasp the strengths and limitations of the model described – a statement similar to that found in the last paragraph of the Materials and methods subsection “Model of phenotypes that contribute to fitness”.

5) A primary result of the study is that mutations that are beneficial in one condition are frequently deleterious in other conditions. This phenomenon of antagonistic pleiotropy has been described frequently in the experimental evolution literature and these prior observations should be more clearly described.

Revisions expected in follow-up work:

Simulations of the structure of genetic modules and their potential to explain these results would be valuable. For example, consider the scenario in which hundreds of "phenotypes" (e.g. the expression of 100 genes) underlies enhanced fitness in the adapted environment, but a change in the environment causes only 10 of those genes to be expressed (i.e. fewer "phenotypes"). What about the converse (10/100)?

---

## [Author Response]

Revisions for this paper:The reviewers took issue with a central claim of the paper, however: a "genotype-phenotype-fitness map" was not constructed as we might normally think, and thus the title is misleading and the goal of such a map was not met. The study does not name and measure the phenotypes themselves; rather it perturbs environmental conditions and measures mutant fitness across environments, which are fitness components that form a collection of abstract phenotypes (as the authors acknowledge) that contribute significantly to fitness.1) The actual phenotypes need to be defined better if we are to accept the title.

We have changed the title to “Fitness variation across subtle environmental perturbations reveals local modularity and global pleiotropy of adaptation” to better reflect the approach we use to identify causal properties of mutations and to avoid using “phenotype” given we don’t directly relate the abstract phenotypes we identify to measured cellular traits.

2) The lack of explicit connection of these fitness components (phenotypes) to known genotypes is also a missed opportunity. Please provide examples of genotype-phenotype relationships in a traditional sense, it will broaden understanding.

We agree that it can be confusing to think about phenotypes in this abstract sense. To try to make this more clear, we’ve added some text to the Introduction. Specifically, we added an example of a more traditional genotype-phenotype inference that one might be able to draw from the mutants in this study. We then describe why this type of inference might not give us all the information about the genotypephenotype-fitness map.

“In the case of the adaptive mutations from Venkataram et al. (Venkataram et al., 2016) mentioned above, we might be able to use our knowledge of the Ras/PKA pathway to make a guess about what phenotypes they affect. […] However, even if these mutations do increase PKA activity, it is not clear how this effect percolates through the system, or what other phenotypic effects we might miss by using such a directed approach to investigate the genotypephenotype-fitness map.”

We also discuss traditional genotype-phenotype relationships in the Discussion section. Here we describe how one might relate the “fitnotypes” we detected to traditionally measured phenotypes:

“Even though fitnotypes are independent with respect to their contribution of fitness and contribute to fitness linearly, the mapping of commonly measured features of cells (e.g. growth rate, the expression levels of growth supporting proteins like ribosomes) onto fitnotypes may not be entirely straightforward. Nonetheless, methods such as Sparse Canonical Correlation Analysis (Suo et al., 2017) hold promise in such a mapping and might help us relate traditional phenotypes to fitnotypes.”

3) One key analysis missing from the paper involves the degree to which the number of phenotypic dimensions is influenced by the number of environments (and mutants). This could be investigated through construction of new models by iteratively removing environments (or mutants) from the existing dataset to see how the number of dimensions (and their predictive power) changes. Inclusion of this analysis is necessary to see how the phenotypic dimensionality is a function of environments (or mutants) tested.

We agree that this is an important set of analyses to understand how our conclusions depend on the number of the environments and mutants used. These analyses will also serve as a guide to future studies that would like to use this approach. We have added two supplementary figures (Figure 4—figure supplements 1 and 2), which show that we detect more components and get better predictive power when we include more subtle environments in the training set or include more mutation types in the training set. We have added accompanying text to the Results:

“The strength of our predictions depends on how many subtle environments we used to generate our phenotype model. […] Randomly excluding many mutation types from the training set decreases our ability to predict fitness, though the exclusion of any one mutation type from the training set has limited impact on our overall predictive accuracy (Figure 4—figure supplement 2).”

In addition, we have added a section in the Materials and methods “Evaluating the effect of the number of environments and mutations used” which contains additional details about this new analysis which may help guide future studies using this approach.

4) In the manuscript, the authors use the term phenotype to represent an abstract entity that contributes to fitness rather than "an observable trait" as it has been classically defined. This distinction should be apparent in the Abstract or Introduction to allow readers to fully grasp the strengths and limitations of the model described – a statement similar to that found in the last paragraph of the Materials and methods subsection “Model of phenotypes that contribute to fitness”.

We agree that this has the potential to be a source of confusion. We have made several changes to emphasize how our use of the term phenotype differs from the definition of observable traits.

We made several changes to the Abstract, which now includes the following sentences:

“We then model the number of phenotypes these mutations collectively influence by decomposing these patterns of fitness variation. […] Importantly, inferred phenotypes that matter little to fitness at or near the evolution condition can matter strongly in distant environments.”

We’ve also added a statement to the Introduction to more clearly signpost what we mean by “phenotype”:

“Importantly, the phenotypes we identify with this approach are abstract entities rather than measured cell properties. Nevertheless, these abstract phenotypes reflect the causal effects of adaptive mutations on fitness.”

We also explicitly highlight that we construct a “genotype-(abstract)phenotype-fitness model” in the last paragraph of the introduction to help the reader understand that our phenotypes are modeled and not measured. In that paragraph, we describe the behavior of adaptive mutations by a “lowdimensional phenotypic model”.

We also introduced text in response to point 2 in the Introduction which will help the reader understand the difference between our approach and an approach that uses the term, “phenotypes,” to refer to traits that are measured via traditional approaches:

“In the case of the adaptive mutations from Venkataram et al. (Venkataram et al., 2016) mentioned above, we might be able to use our knowledge of the Ras/PKA pathway to make a guess about what phenotypes they affect. […] However, even if these mutations do increase PKA activity, it is not clear how this effect percolates through the system, or what other phenotypic effects we might miss by using such a directed approach to investigate the genotypephenotype-fitness map.”

In addition, we describe the difference between our abstract phenotypes v. traditional phenotypes in the Discussion section:

“One disadvantage of our approach is that the phenotypic components that we infer from our fitness measurements are abstract. […] For this reason, perhaps we should not refer to them as phenotypes but rather “fitnotypes” (a mash of the terms “fitness” and “phenotype”) that act much like the causal traits in Fisher’s geometric model…”

5) A primary result of the study is that mutations that are beneficial in one condition are frequently deleterious in other conditions. This phenomenon of antagonistic pleiotropy has been described frequently in the experimental evolution literature and these prior observations should be more clearly described.

We thank the reviewers for reminding us to explain the difference between our observations and observations of antagonistic pleiotropy. This is an important point. We have added the following text to the Introduction to address this:

“In fact the key prediction of this model is that one should be able to detect latent pleiotropy and reveal the additional phenotypic effects of these mutants by demonstrating their varied fitness consequences in other conditions or environments (Figure 1B, right side). […] But it could also indicate that the adaptive mutations all change the same phenotype in a way that improves fitness in some environments and hinders fitness in others.”

Revisions expected in follow-up work:Simulations of the structure of genetic modules and their potential to explain these results would be valuable. For example, consider the scenario in which hundreds of "phenotypes" (e.g. the expression of 100 genes) underlies enhanced fitness in the adapted environment, but a change in the environment causes only 10 of those genes to be expressed (i.e. fewer "phenotypes"). What about the converse (10/100)?

We thank the reviewers for this suggestion and will consider it in follow-up work.